# Traject3d allows label-free identification of distinct co-occurring phenotypes within 3D culture by live imaging

Eva C. Freckmann [1,2,3], Emma Sandilands [1,2,3], Erin Cumming[1,2], Matthew Neilson[2], Alvaro Román-Fernández [1,2], Konstantina Nikolatou [1,2], Marisa Nacke[1,2], Tamsin R. M. Lannagan[2], Ann Hedley [2], David Strachan[2], Mark Salji[2], Jennifer P. Morton[1,2], Lynn McGarry [2], Hing Y. Leung [1,2], Owen J. Sansom [1,2], Crispin J. Miller [1,2] & David M. Bryant [1,2] ✉

Single cell profiling by genetic, proteomic and imaging methods has expanded the ability to identify programmes regulating distinct cell states. The 3-dimensional (3D) culture of cells or tissue fragments provides a system to study how such states contribute to multicellular morphogenesis. Whether cells plated into 3D cultures give rise to a singular phenotype or whether multiple biologically distinct phenotypes arise in parallel is largely unknown due to a lack of tools to detect such heterogeneity. Here we develop Traject3d (Trajectory identification in 3D), a method for identifying heterogeneous states in 3D culture and how these give rise to distinct phenotypes over time, from label-free multi-day time-lapse imaging. We use this to characterise the temporal landscape of morphological states of cancer cell lines, varying in metastatic potential and drug resistance, and use this information to identify drug combinations that inhibit such heterogeneity. Traject3d is therefore an important companion to other single-cell technologies by facilitating real-time identification via live imaging of how distinct states can lead to alternate phenotypes that occur in parallel in 3D culture.

The profiling of cell populations at the single-cell level has transformed quantitative cell biology and unlocked the potential to understand heterogeneous cell states. Markers of distinct cell states, be they genetic, proteomic or morphological features, can be extrapolated to infer cell function[1–7]. Several computational approaches use static time points to predict the sequence in which alternate cell states occur to give rise to alternate phenotypes[8–14]. While powerful, these methods are defined by terminal snapshots, which fail to capture the dynamics of how cell states changing over time is a defining feature of morphogenesis.

The 3-Dimensional (3D) culture of cells or tissue fragments to induce complex multicellular structures, such as cysts, acini, spheroids or organoids, allows in vitro determination of how alternate cell states cooperate to give rise to a phenotype. Static fluorescent imaging of 3D organoids to couple cell morphological features with the spatial distribution of fate or signalling markers has been elegantly used to predict how cell fate changes underpin alternate phenotypes[15,16]. Despite the power of such approaches, most other studies typically rely on the averaging of coarse features, such as size, viability or sphericity, to define changes occurring in response to a treatment. Such simple analyses reflect the combination of increased cost and complexity of sampling 3D volumes over time compared to sampling 2-dimensional (2D) cell populations and a lack of analysis tools for the resulting large datasets. This is a significant bottleneck in realising the potential of 3D culture to identify the extent, repertoire, and biological consequences of heterogeneity.

[1]Institute of Cancer Sciences, University of Glasgow, Glasgow G61 1HQ, United Kingdom. [2]The CRUK Beatson Institute, Glasgow G61 1BD, United Kingdom. [3]These authors contributed equally: Eva C. Freckmann, Emma Sandilands. ✉e-mail: david.bryant@glasgow.ac.uk

Whether 3D phenotypes are largely homogeneous or the extent to which heterogeneity exists in 3D culture is a poorly investigated area. Heterogeneity may represent modest variation in a singular morphogenesis pattern or distinct biological programmes that occur in parallel to result in alternate phenotypes. These programmes may not occur at equal frequencies. For example, a dominant phenotype - defined by high-frequency occurrence of a particular sequence of cell state changes in a cell population - may occur simultaneously with a less-frequent alternate phenotype. It is the numerically dominant phenotype that is most often quantified when using basic 3D culture analyses with low sample numbers. However, numerically rare behaviours may have a disproportionate contribution phenotypically, such as in the case of rare populations that may be metastasis-competent, drug-resistant, or possess stem-like capabilities. 3D culture approaches are increasingly used for drug-response modelling, tissue transplantation and a myriad of other proposed functions. It is essential that the methods for evaluating such cultures are improved such that potentially heterogeneous phenotypes, which may be differentially present in frequency, are considered.

In this work we introduce Trajectory identification in 3D culture (Traject3d), an analysis pipeline that enables detection of heterogeneous phenotypes co-occurring in parallel. Similar to other recent methods[17], we analyse 3D structures from label-free images to identify distinct subtypes co-occurring in heterogeneous populations. However, Traject3d, differs from and improves on this concept by basing phenotype identification on multi-day imaging of objects over time from label-free microscopy. Therefore, unlike approaches that predict how static snapshots might relate in time based on probability of transitions between states (e.g. pseudotime), Traject3d identifies alternate phenotypes based on live-imaging. This opens the door for unbiased identification of potentially rare phenotypes that may occur by unpredicted or low probability transitions between states. We use image segmentation packages CellProfiler[18] and CellProfiler Analyst[19] with downstream analysis performed by Traject3d implemented in KNIME[20] with R[21] and Python integrations. Our software choice is based on accessibility: all are open-source freeware that can be used by biologists without requirement for coding skills. We therefore provide much-needed options for identifying heterogeneity in 3D culture by either user-defined or data-driven methods for biologists.

We use Traject3d to identify how biologically relevant co-occurring phenotypes are associated with enhanced metastatic ability or drug resistance and to identify the genetic and signalling pathways that control these alternate phenotypes. This has enabled us to elucidate a mechanism involving a ligand, its receptor, as well as a downstream effector and its key target. Moreover, we identify that co-targeting this pathway can restore sensitivity to otherwise drug-resistant tumour cells. Although we use Traject3d to identify heterogeneity in 3D in tumour-derived samples, Traject3d can be used on the data from other 3D systems that can be imaged and tracked live over multiple days, in a label-free fashion. Traject3d is therefore an important companion tool to other emerging single cells technologies by unlocking the capacity for data-driven, unbiased detection of co-occurring heterogeneous cell states and how these give rise to alternate phenotypes. We expect Traject3d to therefore be useful for identifying whether heterogeneity exists in a given sample, and a tool to understand the mechanisms by which such heterogeneity may be regulated or contribute to the biological system under study. Without tools such as Traject3d, the contribution of heterogeneous parallel phenotype(s) in biology may continue to be underestimated.

## Results

### Heterogeneity of 3D cultures revealed from live imaging

We aimed to uncover the nature and extent of heterogeneous phenotypes that may occur in parallel in 3D culture. We reasoned that co-occurring phenotypes within a sample could be: i) stochastic variation in a singular 3D phenotype, ii) that seemingly heterogeneous phenotypes may represent a singular phenotype, occurring at different speeds, that is stereotyped but asynchronous, iii) that distinct phenotypes may occur simultaneously, or iv) a combination of these scenarios. In contrast to the common approach of assigning a phenotype from imaging single timepoints, testing these possibilities requires live imaging as potentially heterogeneous phenotypes may occur at different points in time. Similarly, wide parallel sampling is necessary for capturing phenotypes that may occur at disparate frequencies (i.e. capturing rare as well as frequent phenotypes). To aid in the description of our approach we provide a glossary for the definition of key concepts (Supplementary Table 1).

We developed large-scale phase-contrast live-imaging in 3D (Fig. 1a; Methods) by adapting culture methods effectively and extensively utilised to generate highly polarised 3D cultures, including apical-basal polarisation and lumen formation in cell lines and embryonic stem cells, as well as for mechanisms of cancer cell polarisation and invasion into Extracellular Matrix (ECM)[22–32]. This involves a thin coating of ECM applied to 96-well plates, onto which a suspension of single cells is plated in low-percentage ECM-containing medium. In this system, plated single cells undergo division and morphogenesis to become a clonal, multicellular structure polarised around a central lumen. We defined this transition over time as a singular, tracked 'object' (Supplementary Table 1). Use of this 3D culture method fulfilled two purposes: i) the reduction of the amount of ECM used, making the assay more cost-effective, and ii) the formation of multiple 3D multicellular structures undergoing morphogenesis in a largely similar plane and field of view. This facilitated multiday imaging using autofocus approaches and scalability to image hundreds of thousands of spheroids across multiple 96-well plates in parallel using the Incu-Cyte system. We used CellProfiler for image segmentation[18] (Fig. 1a) and tracking of 3D structures over multiple days per treatment condition. This approach facilitates analysis of hundreds of thousands of objects in an experiment (Supplementary Tables 2, 3), enabling interrogation of potential co-occurring phenotypes across time with robust statistical support.

We tested this approach by imaging a broad range of normal and tumour-derived human and mouse cell lines as they formed 3D spheroids over multiple days (>1.6 million objects), measuring size, shape and movement features of objects (Supplementary Table 2; Supplementary Figs. 1–3; see Traject3d GitHub for CellProfiler segmentation and feature measurement pipelines). Visual inspection of time-lapse imaging revealed instances of phenotypic homogeneity across time in spheroid development (e.g. immortalised, non-transformed RWPE-1 normal human prostate cells); in a number of samples, distinct spheroid phenotypes occurred in parallel within a sample across time (e.g. metastatic PC3 human prostate cancer cells) (Supplementary Figs. 2a, b and 3a). To assess the repeatability of such global behaviours, we compared mean shape, size and movement features by Principal Component Analysis (PCA) from samples trimmed to the same imaging length, to ensure appropriate temporal comparison. This indicated that the size, shape and movement features in a given sample were largely concordant across both intra-experiment technical replicates and across independent experiments (Supplementary Figs. 2c, d and 3b, c). This was further supported by the visualisation of phenotypic space across experimental replicates through t-distributed Stochastic Neighbour Embedding (t-SNE) of shape, size and movement features across time (Supplementary Fig. 2e, f). This suggested that phenotypic heterogeneity was unlikely to be solely from stochastic variation.

PCA analysis also enabled the identification of instances of batch effects (dashed lines) as well as segregation of samples with mostly spherical, poorly motile phenotypes (MDCK, RWPE-2, CWR, 22Rv1, Caco-2) from mostly elongated, motile phenotypes (MDA-MB-231,

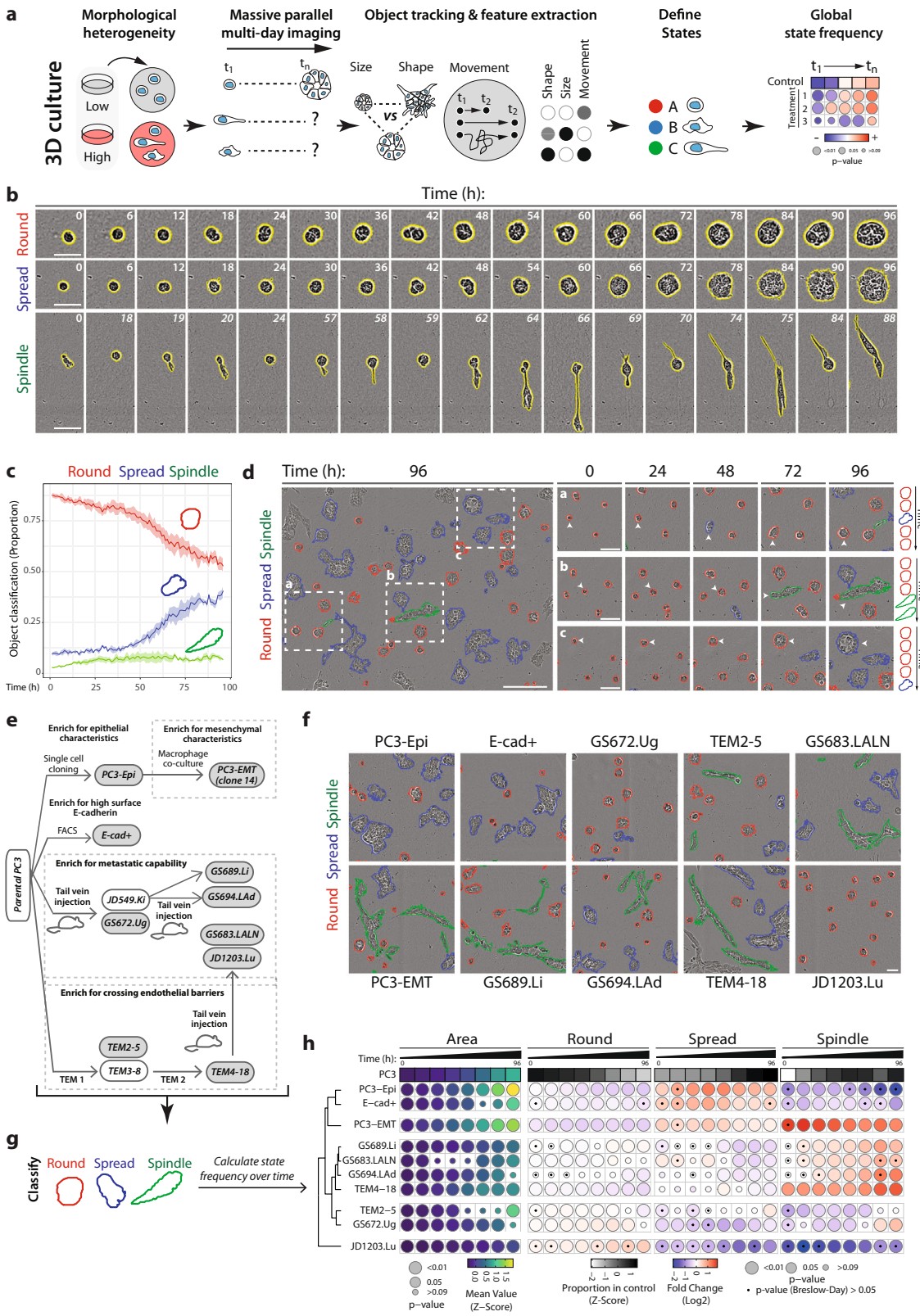

PC3M, PC3M-D^R) (Supplementary Figs. 2c and 3b). However, this failed to capture that some samples display co-occurring distinct phenotypes (both round and elongated in MDA-MB-231 and PC3M-D^R). Therefore, additional approaches are needed to identify distinct phenotypes occurring within a sample, potentially at a lower frequency, without which functions like PCA might otherwise skew sample analysis towards the most frequent behaviour(s).

## User-defined classification of heterogeneous states

We examined two approaches to identify co-occurring phenotypes within 3D cultures: the application of user-defined classifications or a data-driven unbiased subtype detection. In the first approach, to classify spheroids into user-defined groups we required a simple click-and-classify methodology that i) allowed classification with high efficiency, and ii) contained a machine learning model that, after training,

**Fig. 1 | Alternate phenotypes occur in parallel within 3D cultures. a** Schema, heterogeneous spheroids imaged in high-throughput over time. Size, shape and movement characteristics extracted for thousands of spheroids. Machine learning used to classify user-defined phenotypic states, frequency of which was quantified over time. **b** Representative phase images of spheroids exhibiting variable morphology over time. $n = 3$ independent experiments, 3 wells/condition/experiment. Scale bars, 50 µm. **c** Proportion of PC3 spheroids exhibiting user-defined classification states. Shaded region, s.e.m. across experiments. $n = 3$ independent experiments, 3 replicates/condition/experiment. Total number of spheroids quantified in Supplementary Table 3. **d** Representative phase images of spheroids. Outlines, user-defined state classification. Scale bar, 100 µm. Time-lapse of boxed regions shown. Arrowheads and schema indicate changes in classification over time. Scale bars, 50 µm. $n = 3$ independent experiments, 3 wells/condition/experiment. **e** Schema of PC3 subline derivation. PC3 were selected in vitro for epithelial shape (PC3-Epi), high surface E-Cadherin (E-cad+) or mesenchymal characteristics after macrophage co-culture (PC3-EMT). PC3 were injected into murine tail veins and harvested from alternate metastatic sites; GS689.Li (liver), GS672.Ug

(urogenital tract) and GS694.Lad (adrenal gland, after in vivo injection of PC3 JD549.Ki). Sublines were isolated after serial passage across endothelial barriers (TEM2-5 vs TEM4-18). TEM4-18 were injected into tail veins and cells harvested from lymph node (GS683.LALN) and lung mets (JD1203.Lu). **f** Representative phase images of PC3 subline spheroids, 72 h. Outlines, user-defined state classification. $n = 3$ independent experiments, 3 wells/condition/experiment. Scale bar, 100 µm. **g** Representative outlines of phenotypes formed by PC3 sublines. **h** Quantitation of PC3 and sublines. Heatmap shows Area as mean of Z-score normalised values (purple to yellow), and classification into Round, Spread or Spindle as a Log2 Fold Change from control (PC3) (blue to red). Proportion of control at each timepoint is also Z-score normalised (white to black). Bubble size represents p-values, Student's t-test (two-sided) and Cochran-Mantel-Haenszel test, Bonferroni adjusted, to compare Area and proportion of each classification to control respectively. Dot represents *p*-value, Breslow-Day test, Bonferroni-adjusted for homogeneity of odds ratio across experimental replicates. $n = 3$ independent experiments, 3 wells/condition/experiment. Number of spheroids quantified in Supplementary Table 3.

could be converted into 'rules' that were exported and applied to subsequent data sets. The key here is to enable a user to apply consistent classification to each new data set as it was acquired rather than wait until all data collection was complete before generating a classification model, an approach that while valid is not always practical in a laboratory setting. The Fast Gentle Boosting machine learning model in CellProfiler Analyst[19] met these criteria, enabling us to classify spheroids into user-defined states (Fig. 1a).

In contrast to the largely homogeneous RWPE-1 spheroids, PC3 spheroids could be classified into three categories (based on extensive visual examination of time-lapse movies) with high fidelity to the true user classification (91–97%; Fig. 1b-d; Supplementary Figs. 2a and 4a–c): objects that remained round ('Round'), those that locally spread ('Spread') and those displaying an elongated shape ('Spindle'). To aid in visualisation of these classifications we computationally selected a representative outline of each state and identified the measurements of area, shape, and motility that define them (Supplementary Fig. 4d–f; Methods). Such heterogeneity could also be observed from maximum projections of sequential optical sections via confocal imaging of PC3 spheroids (Supplementary Fig. 5a, b).

Applying user-defined classifications to multiday imaging revealed that the relative proportions of distinct states in a sample vary over time in a strikingly consistent fashion across independent experiments (Fig. 1c). Applying state classification to still frames from live imaging revealed a remarkable plasticity to cell state, wherein distinct states are neither static nor disconnected. This diverges from previous approaches which use phenotype classification from limited timepoints[33–37] that might be extrapolated to assume that states are either static or transitions are unidirectional. Although most objects start as round, objects can move between state classifications over time, both at different times and rates, and near noncontacting spheroids undergoing alternate distinct behaviours (Fig. 1b; Fig. 1d, arrowheads). This indicates that state classification without application longitudinally to live imaging likely underestimates state oscillations that may lead to distinct phenotypes.

A key question in identifying heterogeneous phenotypes is whether distinct states are associated with alternate behaviours. If so, then repeated or independent selection for these behaviours should converge on the same cell states. We tested this using a series of existing cell lines that were independently derived for altered invasive or metastatic abilities and examining their temporal state changes. We compared independent subclone derivation from the heterogeneous parental PC3 cells (Fig. 1b-d) for, i) in vitro selection for epithelioid characteristics (PC3-Epi[38]) or high surface E-cadherin expression (E-cad+[39]), ii) in vitro selection for mesenchymal characteristics after initial co-culture of PC3-Epi with macrophages, then isolation of the resultant PC3-derived cells (PC3-EMT[38]), iii) in vitro enrichment for

trans-endothelial migration (TEM2-5, TEM4-18[39]), and iv) in vivo selection for metastasis by harvesting from alternate metastatic sites after tail vein injections without (GS672.Ug, GS689.Li, GS694.LAd[39]), or with (GS683.LALN, JD1203.Lu[39]), prior in vitro trans-endothelial migration (Fig. 1e).

Imaging of spheroids from these PC3-derived lines (Fig. 1f), and from other cell lines (Supplementary Figs. 2 and 3), highlighted that heterogeneous behaviours can alter analysis in complex ways, such as that while spherical non-motile spheroids stayed distinct throughout multi-day imaging, highly motile or elongated phenotypes could eventually cause spheroid merging, at late time points. We therefore limited our analysis to a time interval where the majority of single spheroids could be appropriately detected as individual objects across all samples being compared – in the case of PC3 cells and its derivatives, generally 96 h.

We quantified over time the size (Area) of objects, the proportion of objects classified into user-defined categories in the parental PC3 (Round, Spread, Spindle), and the relative fold-change in these proportions of each subline compared to the parental (Fig. 1f–h) (>1.2 million objects, 4 days of imaging: Supplementary Table 2). For ease of visualisation of multiple timepoints (~96 h) we condensed changes into mean phenotype (Area, state) in 12-hour intervals (Fig. 1h). Statistical comparison of changes was analysed through a Cochran-Mantel-Haenszel test, wherein statistical significance is only achieved where an effect was consistent across independent experiments. Furthermore, we use the Breslow-Day test to assess the differential magnitude of effect across biological replicates. In the heatmap, a non-significant value indicating consistent magnitude of effect is represented by a black dot. Therefore, these heat maps depictions of change across time represent not only the change in a parameter in relation to the control sample, but also the significance and consistency across repeated experiments.

We observed three main trends in these independently generated cell lines, wherein one classification type was enriched at the expense of the other two: i) the Spread state was largely selected in those with epithelial characteristics (PC3-Epi and E-cad+), ii) Spindle state in a subset selected from metastases (GS689.Li, GS683.LALN, GS694.LAd) or two rounds of in vitro selection for trans-endothelial migration (TEM4-18), or iii) modest increase in Round state from other metastatic samples or a single round of trans-endothelial migration (TEM2-5, GS672.Ug, JD1203.Lu). Notably, when clustered by Euclidian distance, due to differences in the strength of the trend, JD1203.Lu was distanced from the other lines, exhibiting an increase in Round state. The PC3-EMT sample was the only instance of induction of dual behaviours (Spread and Spindle). This reveals that independent selection of sublines for a certain behaviour (e.g. metastasis or invasion) converges on similar phenotypic characteristics. This suggests that heterogeneity

represents functionally different co-occurring phenotypes. Ensuring accurate detection of the repertoire of phenotypes is therefore essential, as rare phenotypes may include those that have special characteristics, such as metastasis competency. This approach can therefore be used to identify distinct 3D behaviours within a sample and assess the consistency of behaviours across time and independent experimental replicates.

## Data-driven identification of heterogeneous states by Traject3d

An open, essential question is how many cell states may co-occur in a sample. For instance, is elevation of both Spread and Spindle states in PC3-EMT cells (Fig. 1h) an enrichment of two independent phenotypes or a phenotype that that oscillates between these states, or a de novo state that resembles both classifications and is consequently poorly classified? The application of user-defined states, while powerful for identifying specific phenotypes of interest, suffers from limitations; these include a requirement for manual training, an unknown depth limit to which a user can manually identify states, the use of static images preventing classification based on motility features, and forced classification into user-specified classes regardless of fit. To overcome some of these limitations and address such questions, we developed a method of data-driven subtype classification and analysis of behaviour patterns over time to identify unique events.

To identify distinct states in a data-driven manner, thousands of objects tracked from time-lapse images need to be analysed for size, shape and movement features across what may be a hundred, or more, time points (Fig. 2a). To reduce computation time we calculated the cross-correlation of these features across the 22 cell lines previously analysed (Supplementary Table 2, Supplementary Fig. 1) and manually identified a subset of non-redundant features (Supplementary Table 4). Size is defined by Area, shape via Zernike polynomials, and movement features generated by CellProfiler (Displacement, Distance Travelled, Integrated Distance and Linearity). We did not use texture features as these are affected by variations in the focal plane of imaging. Using these non-redundant features we collectively analysed data from samples and their controls for remaining analyses (Supplementary Table 3) allowing direct comparison of phenotypes. As discussed previously, a key point is that the above samples were similar enough in their morphogenesis characteristics that image sequences of the same length were compatible.

This multi-dimensional data was used to identify distinct states using PhenoGraph[40], an approach common for such datasets in mass cytometry (Fig. 2a). We selected this over other algorithms (ClusterX[41], DensVM[42], FlowSOM[43], k-means[44]) as it allowed unsupervised clustering into subpopulations without reliance on prior dimensionality reduction (such as t-SNE). To reduce computation time, we subsampled 20,000 objects evenly across phenotypic space using GeoSketch[45], then fitted the remaining data to the identified states (see Methods). A challenge in identifying distinct states is generating a meaningful label for each subtype; in single-cell sequencing this can be done by mapping expression profiles onto reference datasets to identify if they represent known cell types. This does not exist for analysis of time-lapse 3D imaging.

To aid in visualisation and interpretation, we generated a method for computationally selecting representative outlines for each identified state (Supplementary Fig. 6; Methods). Notably, some phenotypes that an expert user can distinguish as distinct can nonetheless result in objects with similar features, such as some non-round states and instances of spheroids merging. As the underlying features are highly similar, in a small number of cases these can be classified as the same state (see state K in Supplementary Fig. 6). In a perfect experimental system these events would not occur. However, they are bona-fide occurrences likely to be observed in most methods utilising 3D cultures in vitro. Therefore, Traject3d provides an honest capture of the reality of 3D culture and allows the user to interpret biological significance,

rather than pre-emptively excluding data and introducing bias. We visualised the data using t-SNE (Fig. 2a,b)[46–48], which performed superiorly to PCA but equivalently to UMAP[37] (Supplementary Fig. 7a,b)[49,50].

We quantified the identified states to determine enrichment or depletion relative to the control (Fig. 2a). In contrast to expert user definition of three states (Fig. 1), a data-driven approach identified sixteen states occupying defined regions in phenotypic space (State A-P; Fig. 2b, c), that occurred with remarkable consistency across time and independent experimental replicates (Fig. 2d). Comparison of this approach to user-defined classifications revealed subdivision both largely within (e.g. state C within Round), as well as borderline between (e.g. state I across Spread and Spindle), user-defined state classifications (Fig. 2b, c). Comparison of the relative proportion of objects within each classification revealed six states with highest frequency in parental PC3 cells: states C, H, O (largely Round), state I (borderline Spindle/Spread), and states K, M (borderline Round/Spread) (Fig. 2d). These included states that were frequent but decreased over time (state C), were somewhat constant (states H, K), or increased over time (states I, M). In addition, ten rare (<5%) states were observed, with some (states G, L, P) increasing in frequency at later time points. A data-driven approach to state classifications can therefore provide clarity to regions where user-defined classifications perform poorly.

Having a granular analysis of state (sixteen states) allowed data-driven clustering of independently derived PC3 sub-lines and the states that define them. We quantified global state frequency relative to control (parental PC3 cells; Fig. 2e) and used PCA of size, shape and movement features to visualise the relationships between states (Supplementary Fig. 8a, b). This revealed three broad groupings of states defined by: largest size (L, D), a lack of motility (E, C, K, H, O, G, J, M), or possessing motility (B, F, P, I, N, A) (Fig. 2b, c, e). Similar to user-defined states, statistical comparison to control was performed using Cochran-Mantel-Haenszel and Woolf tests to take into account the consistency and magnitude of change across experimental replicates.

Rather than adopting a singular shape, the epithelioid group (PC3-Epi, E-cad+) shared a range of shapes from round to modest elongation that were all defined by a lack of motility (K, H, O, G, J, M; Fig. 2e). A second group of cells (GS689.Li, GS694.LAd, GS683.LALN, PC3-EMT, TEM4-18) were largely defined by motile, enlarging, elongated cells (states B, F, P, I, N). The third group of cells (TEM2-5, GS672.Ug) were defined by modest changes from the parental PC3 cells, while JD1203.Lu possessed a large switch to two predominant states (states E, C) that lack motility. Notably, shape alone was insufficient to define behaviours as a small, round state (state A) possessed enhanced motility and was one of the features of the metastatic cells. Similarly, cell elongation (state E) was a feature of both epithelioid cells (Group A) and metastatic cells (Group B) as it occurred with an absence of motility. These data indicate that both user-defined and data-driven analysis of heterogeneity can be used to independently detect states occurring in 3D culture, even those at low frequency. However, only data-driven approaches provided the granularity and incorporation of motility features required to detect the full repertoire of cell states.

## Identification of distinct states leading to alternate morphogenesis patterns

An important feature gleaned from live imaging is that behaviour in 3D is highly dynamic. This suggests that in addition to identifying the repertoire of cell states, the order in which they occur across time needs to be considered to uncover how alternate states lead to alternate phenotypes. Static imaging or sequence prediction from limited observations will likely underestimate the complexity of state changes that can define a phenotype, emphasising that timelapse imaging is key to understanding distinct phenotypes.

A challenge in basing analysis on 3D objects over time is that a) objects must remain as single objects during observation and b) that objects with different motility characteristics may not be equally

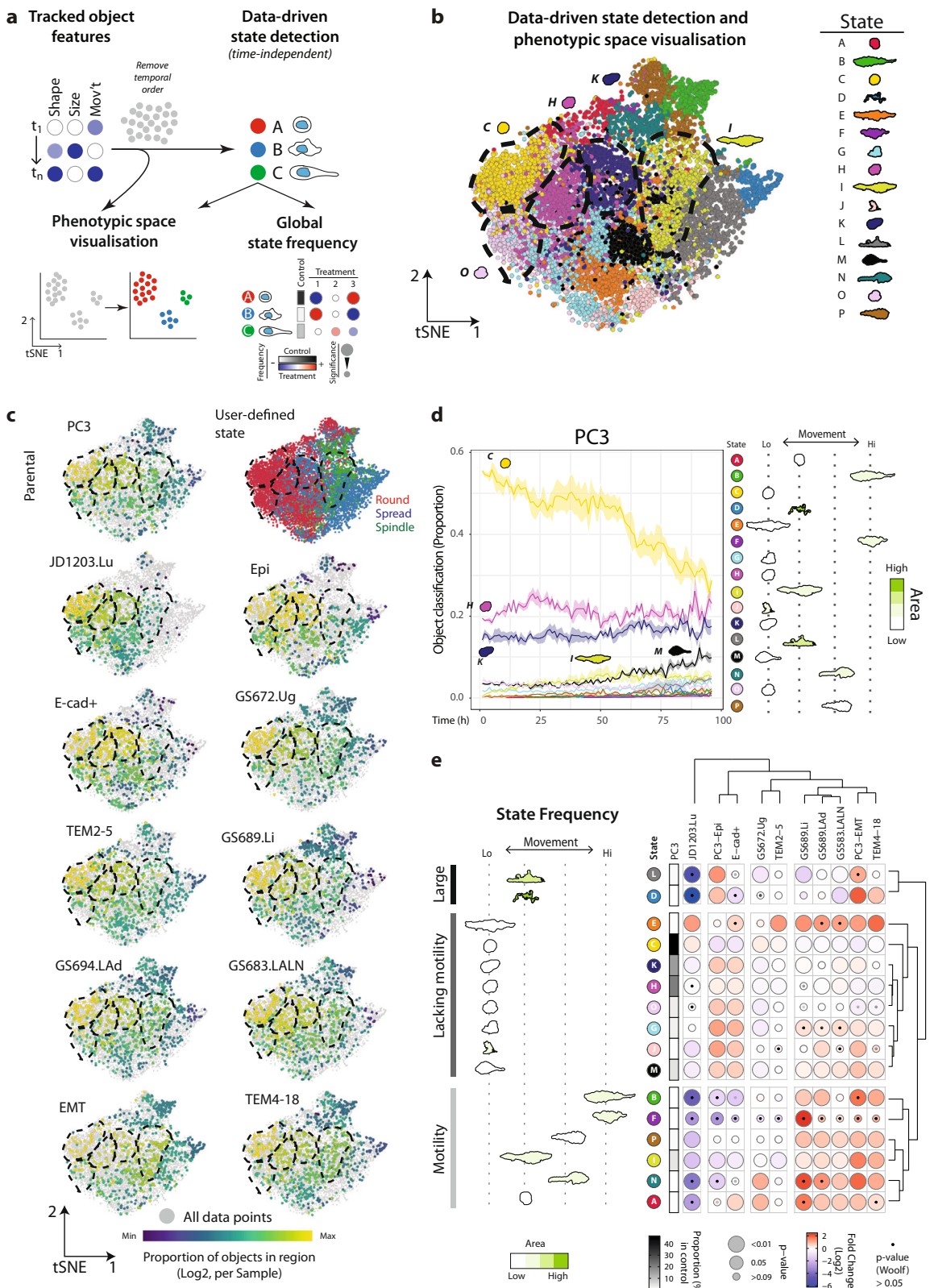

tracked using a singular tracking algorithm. For instance, multiple unique spheroids may touch in one frame of an image sequence or split into daughter objects in another frame. This is a challenge for analysis as two tracked spheroids can share a common event (aggregate object). To partially address this we applied a tracking label correction at points of splitting/merging where the largest daughter object retains the label of the parent; other child objects are assigned new unique labels (see Methods for more detail). This allows for retaining at least one of the

objects rather than discarding what could otherwise be days of tracking. Such instances are then represented by the appearance of cell states enriched for touching or splitting (e.g. cell state K). This allows bias-free capture of the events in 3D culture and the option for user interpretation of the biological significance of the events.

As distinct cell states are identified in data stripped of time, cell state is agnostic of the tracking labels. However, some behaviours may include motility that is faster than the imaging interval frequency.

**Fig. 2 | Identifying the repertoire of distinct states that occur in 3D cultures.**
**a** Schema, analysis of heterogeneous spheroids, regardless of temporal order,
enables data-driven subtype classification, visualisation of phenotypic space, and
frequency relative to control. **b** t-SNE of states in PC3 and sublines. Plot points
coloured by data-driven state classification. Black dashed lines highlight regions
corresponding to data-driven states mentioned in text. Data comprised of each
spheroid identified in each image of the experiment. Number of spheroids quan-
tified in Supplementary Table 3. t-SNE analysis performed on 20,000 objects
subsampled via GeoSketch, with iterations = 2000, theta = 0.5, perplexity = 50.
**c** t-SNE of the relative distribution of spheroids for PC3 and sublines. Plot points
coloured by user-defined state classifications, and data-driven state enrichment:
purple-to-yellow shows per sample proportion of total objects in each data-driven
state, quantified before t-SNE. Black dashed lines, highlight regions corresponding
to data-driven states mentioned in text. Data comprised of each spheroid identified
in each image of the experiment. Number of spheroids quantified in Supplementary
Table 3. t-SNE analysis performed on 20,000 objects subsampled via GeoSketch,
with iterations = 2000, theta = 0.5, perplexity = 50. **d** Proportion of PC3 spheroids
exhibiting each data-driven state classification over time. Shaded region represents
s.e.m. across experiments. n = 3 independent experiments with 3 wells/condition/
experiment. Number of spheroids quantified in Supplementary Table 3. Repre-
sentative outlines shown, arranged by average movement, and coloured by average
size (green scale). **e** Quantitation of data-driven state classifications in PC3 subline
pairs. Representative outlines shown, arranged and coloured (light to dark green)
by average class motility and size, respectively. Heatmap shows classification of
spheroids as a Log2 Fold Change from control (PC3) (blue to red). Proportion of
control in each class is shown (white to black). Bubble size represents p values,
Cochran-Mantel-Haenszel test (Bonferroni-adjusted), to compare each classifica-
tion to control. Black dot represents p-value, Woolf test (Bonferroni-adjusted), for
homogeneity of odds ratio across experiments. n = 3 independent experiments, 3
wells/condition/experiment, number of spheroids quantified in Supplementary
Table 3.

While most features used to define cell states (size, shape) are unaf-
fected there may be some contribution of imperfect tracking to
motility features resulting in some underestimation of motility con-
tribution to some cell states. Therefore, while Traject3d can analyse
the data generated from image segmentation and tracking of time-
lapse imaging, a key consideration is that any analysis will be restricted
by the limitations above, some of which are inherent to 3D culture
analysis. We therefore may detect many, but not all types of beha-
viours. Nonetheless, Traject3d clearly identified behaviours that were
distinguished by their motility features (Fig. 2e; Supplementary Fig. 9).

While computational approaches exist that predict the temporal
order of state transitions from a limited set of static timepoints[8–14],
Traject3d uses true temporal ordering of individually tracked spher-
oids from live imaging. In brief, we restored the temporal order of the
assigned data-driven state classifications and filtered the tracked
spheroids retaining only those which i) existed from the beginning of
imaging and ii) were tracked for a length of time compatible with
comparison (see Methods). In the instances of spheroid merging or
splitting the size and shape of the new aggregate object has changed
significantly such that CellProfiler assigns it a new unique tracking
label, which is inherited by the daughter objects. These 'new' objects,
as defined by the assignment of a new tracking label, do not exist long
enough for inclusion in the comparison of patterns, and therefore do
not contribute to the identification of trajectories. As a result, and
accounting for the above considerations in tracking distinct pheno-
types over time in 3D culture, we were able to analyse the temporal
sequences of 18,922 spheroids tracked every hour over four days
(Supplementary Table 5). This allowed us to analyse recurring tem-
poral patterns of state changes across time that lead to each pheno-
type (Fig. 3a).

Traject3d identified twenty-three distinct patterns of state
change, which we term 'trajectories' (Supplementary Fig. 10). Each
pattern is summarised by a motif that represents the frequency of
state(s) across time, accompanied by the representative outlines of the
most frequent state in each time period (Fig. 3a, Supplementary
Fig. 11). To aid in the interpretation of each trajectory classification, a
representative spheroid corresponding to the sequence of the most
frequent state, with matching pseudo-coloured outline, is provided
(Fig. 3a–e; Supplementary Fig. 11). This was computationally selected
in an unbiased manner, by selection of a spheroid sequence over time
whose temporal state classifications most closely matched its trajec-
tory group's sequence of most frequent state across time (see Meth-
ods). Although some trajectories appear similar when binned into
fewer time periods, such as having the same state as the most frequent
(e.g. state C), they differ by distinct patterns of transitions between
states over time (Fig. 3a, Supplementary Fig. 12). This illustrates the
fact that distinct phenotypes arise by either using distinct cell states or
by using the same states in a different order over time. Animations of

these changes per trajectory are provided as Supplementary Videos 1-
23. The quantities of identified states and trajectories were robust
against changes in the parameters used to identify state and trajectory
clustering (k-nearest neighbour parameter; Supplementary Fig. 13,
Supplementary Note 1).

As shown previously for the quantification of user-defined and
data-driven states, we compared enrichment or depletion of trajec-
tories relative to control using statistical tests (Cochran-Mantel-
Haenszel and Woolf) that consider consistency across experimental
replicates. Parental PC3 displayed most trajectories (eighteen) simul-
taneously with a frequency of 2-15%, as well as five rarely (<2%)
occurring trajectories (Fig. 3b-e). Traject3d identified and grouped
remarkably consistent trajectories between the two independently
isolated epithelioid lines (Group I, pink; PC3-Epi, E-cad + ), versus
highly invasive/metastatic cells (Group II, brown; PC3-EMT, TEM4-18,
GS689.Li, GS694.LAd, GS683.LALN) or in those with more minor
invasion (Group III, purple; JD1203.Lu, GS672.Ug, TEM2-5). Cell group-
specific trajectories could be observed, such as: the largely round
trajectory 6 for epithelioid cells (Group I; Fig. 3c), trajectory 7 that
represents motile objects enriched in state I (elongated state; Fig. 3d)
in Group II, or a lack of both trajectories for Group III (Fig. 3b, green
highlighted box). Instances where cell groups differed by a single tra-
jectory could also be identified (e.g. trajectory 21 between PC3-Epi and
E-cad + ; Fig. 3b, green highlighted box).

Each cell group was not defined by alteration of a singular tra-
jectory, but by a change in several co-occurring phenotypes (e.g.
decrease in trajectories 8/13/15/18 and increase in trajectories 19/16/6/
10 in epithelioid group I; Fig. 3b). Although Trajectories 6 and 7 are
differentially frequent between cell groups, both trajectories share a
large fraction of common states (states C, H, K) and have a similar
repertoire of state transition profiles (Fig. 3c, d). These differ in that the
epithelioid-enriched trajectory 6 mostly remains in the common,
poorly motile states (states C, H, K), while trajectory 7 invokes rarer (I,
M) states at later time points. This emphasises that only imaging over
time to identify the patterns of change, rather than static imaging to
assign a phenotype, could distinguish between phenotypes with
similar states.

The similarity between clustering of the data based on their state
and trajectory classifications is worth noting, although it is expected.
The trajectories are built from the cell states, ordered in sequence
from live imaging. However, a deconstructed view of the data cannot
alone capture how states may transition from one to another over time
to form distinct phenotypes. While clustering of these cell lines by
their user-defined (Fig. 1h) and data-driven (Fig. 2e) states generally
resulted in similar groupings, only analysis of the same data *over time*
enabled grouping of the former outlier, lung metastasis-derived
JD1203.Lu, into Group III. Moreover, this analysis uncovers the time
points at which trajectories differ, which can occur at distinct time

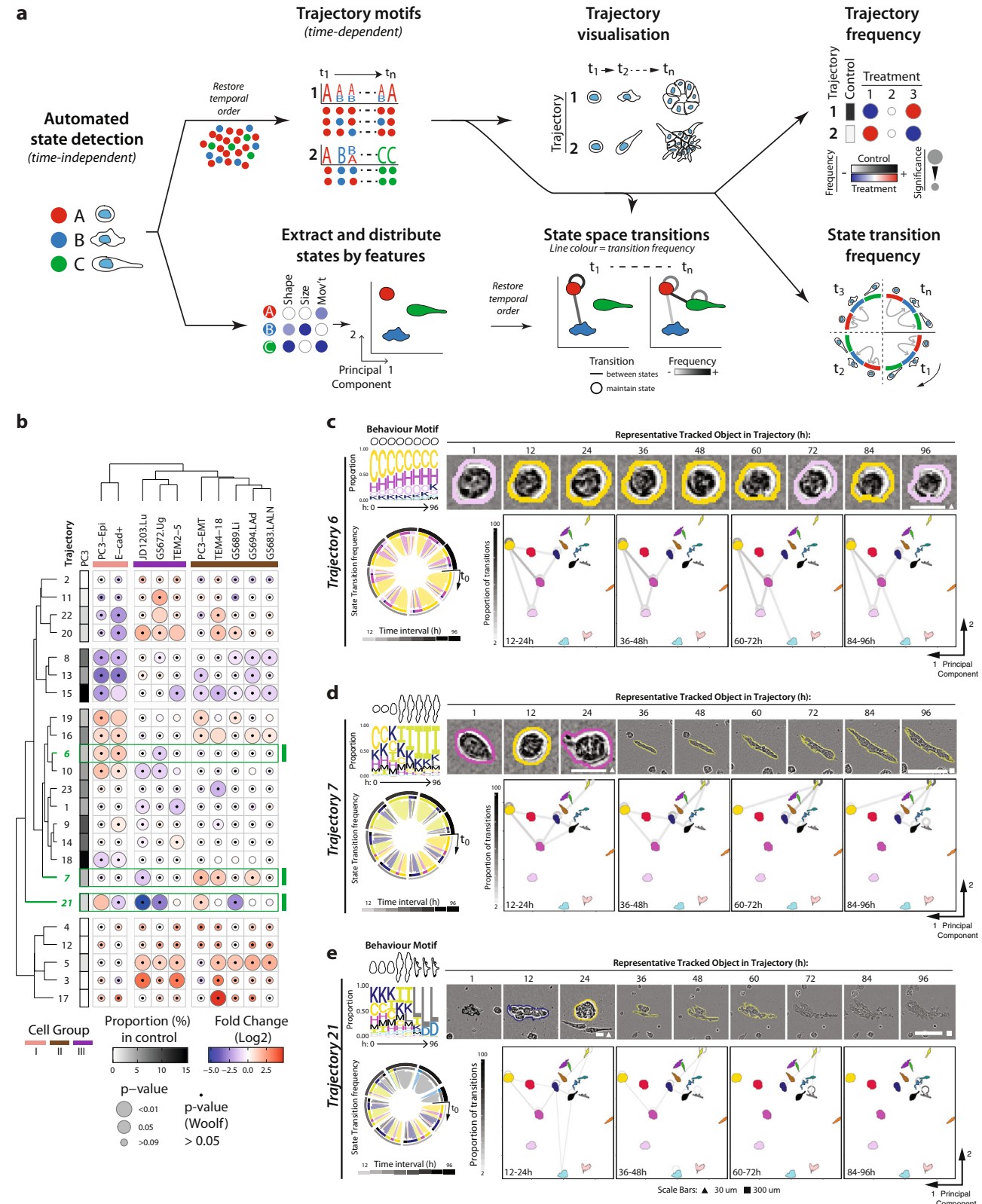

points between different trajectories under comparison. This emphasises that analysis from a single timepoint, or without consideration of time, fails to elucidate the true repertoire and order of heterogeneity. This shows that Traject3d can robustly identify distinct phenotypes in a data-driven fashion, facilitating identification of behaviours associated with a particular biological function such as epithelioid organisation or metastatic activity.

**Traject3d allows deconvolution of bulk sequencing to identify morphogenesis pathway regulators**

Traject3d identifies co-occurring phenotypes within a sample, and whether these phenotypes differ between samples. We reasoned that this may allow deconvolution of bulk RNAseq, as changes in gene expression profiles that underpin each phenotype should be altered in a similar trend to the phenotypes themselves;

**Fig. 3 | Data-driven identification of distinct phenotypes occurring in parallel over time. a** Schema, illustrating analysis of temporal data to determine morphogenesis trajectories. Analysis of size, shape and movement characteristics, regardless of temporal order, enables classification of data-driven states. These states can be visualised by distributing them in 2-dimensional space based on mean features. By restoring temporal ordering of tracked spheroids, a sequence of state events for each spheroid can be generated. Recurring trajectories of state change over time are determined from this, and summarised as a state frequency motif over time. Using the most frequent state at each timepoint, a spheroid is selected to represent each trajectory. Transitions between states are quantified, before visualisation projected onto state space and as a chord diagram. **b** Heatmap shows quantitation of trajectory classification of spheroids as a Log2 Fold Change from control (PC3) (blue to red). Proportion of control in each trajectory is shown (white to black). Bubble size represents p-values (Cochran-Mantel-Haenszel test) comparing each classification to control. Black dots represent *p* value (Woolf test)

testing homogeneity of odds ratio across experimental replicates. *n* = 3 independent experiments, 3 wells/condition/experiment. Spheroids quantified, after filtering, in Supplementary Table 5. Trajectories discussed in text, highlighted in green. Trajectories discussed in text highlighted in green. Cell Groups (I, pink; II, brown; III, purple) derived from dendrogram. **c–e** Trajectory visualisation. Colours represent previously identified states and correspond to those used in Fig. 2b. Behaviour motif depicting frequency (proportion) of states in 12-hour time intervals, with outline of the most abundant state shown at top. Using the most frequent state at each timepoint, a spheroid was selected to represent the trajectory, phase images shown, outline colour indicating state at given timepoint. Scale bars, 30 μm (triangle) and 300μm (square). Transitions between states shown globally as a chord diagram, with time interval in greyscale. PCA used to arrange states in 2-dimensional space; transitions (shown as proportion; greyscale) between (lines) and maintaining (circles) states are overlaid onto this for select time intervals.

Traject3d identifies which cell lines to pair to perform such comparisons.

Using this approach, we selected two pairings that represent one cell line per pair from each of epithelioid (Group I) versus invasive (Group II) phenotype. Cell Pair 1 (PC3-Epi versus PC3-EMT) resulted from direct derivation: co-culture of cells cloned for epithelial phenotype (PC3-Epi) with macrophages in vitro before re-isolation of the epithelial cells alone (PC3-EMT[38]) (Fig. 1f). For Cell Pair 2, cells were derived independently: sorting for high levels of surface E-cadherin (E-cad + ), compared to cells that metastasise to liver (GS689.Li[39]). This ensured that we avoid derivation-specific effects, such as the induction of trajectory 21 specific to Pair 1's derivation (Fig. 3b, e; green highlighted box).

We mined existing RNAseq experiments of these pairs for differential expression and pathway activity (Fig. 4)[38,39]. Known epithelial identity genes were amongst the highest transcripts expressed in cells in Group I from the cell pairings (PC3-Epi, E-cad + ) (Fig. 4a, b). As an Epithelial-Mesenchymal Transition (EMT) was apparent in the invasive cells from both cell pairings (PC3-EMT, GS689.Li; Fig. 4c), we queried the changes in EMT-related transcriptional regulators. This revealed that the master transcriptional regulator of EMT, ZEB1, was strongly induced in invasive cells[51] (Fig. 4d). We calculated the Top 50 epithelioid or EMT-associated genes, defined as those with highest differential expression in common between epithelioid (PC3-Epi, PC3-E-cad + ) versus mesenchymal (PC3-EMT, GS698.Li) cell pairings (Fig. 4e). As ZEB1 is typically a repressive transcription factor of genes associated with epithelial identity, we compared the identified 'Top 50 epithelioid genes' to RNAseq data from control *versus* ZEB1-depleted (two independent clones) mesenchymal PC3-EMT cells. This revealed that 20% of the 'Top 50 epithelioid genes' had their expression substantially restored (often 10-fold on a Log2 scale) after expression of *ZEB1* shRNA (red, asterisks; Fig. 4e). Conversely, the cell lines selected for epithelial characteristics (PC3-Epi) or high E-cadherin expression (E-cad + ) in either pair lacked ZEB1 expression and displayed robust levels of the ZEB1-repressed target gene E-cadherin and the master regulators of epithelial-specific splicing patterns ESRP1/2[52] (Fig. 4e, f). This suggests that the balance between epithelioid and invasive state trajectories within each cell line may be controlled by a ZEB1-driven EMT.

We reasoned that if we were able to use the phenotypic changes detected by Traject3d to deconvolve molecular regulators from bulk RNAseq, then introducing such alterations in parental PC3 cells should recapitulate the same phenotypic changes. Indeed, ZEB1 or ESRP1/2 depletion in parental PC3 (Supplementary Fig. 14a, b) resulted in a significant switch between the broad user-defined states of Round and Spindle (Fig. 5a-d). This initially suggested that the ZEB1-ESRP1/2 axis may function to control cell shape. However, the data-driven approach revealed that while ZEB1 depletion did affect predominantly elongated cell states, ZEB1 was not a regulator of cell shape per se. Rather, ZEB1

depletion broadly repressed states with increased motility (states L, D, B, P, I, N; Fig. 5e, f) irrespective of their shape (note state A is round but motile), while increasing the prevalence of elongated, but poorly motile states (states E, M; Fig. 5e, f). This led to a depletion of invasive trajectory 7, which contained these motile states (e.g. state I) (Fig. 5e-j; see Fig. 3d for trajectory summary). Paradoxically, ZEB1 depletion also decreased the highly round trajectory 3 (Fig. 5h, j), concomitant with the enrichment of several trajectories containing varied cell shape, but all lacking motility (trajectories 19, 6 10, 23, 1, 18, 5) (Fig. 5f, h, k). This reveals an unexpected function of ZEB1 in the adoption of motility rather than cell shape; in the absence of ZEB1 cells move out of a round state (e.g. state C, and leading to Trajectory 3), but become unable to adopt motility features, instead displaying a range of alternate shape states that ultimately leads to multiple alternate phenotypes (trajectories 19, 6 10, 23, 1, 18, 5) (Fig. 5f, h, k).

Depletion of ESRP1/2 resulted in mostly Spindle-type user-defined classification (Fig. 5c, d), which suggested that these splicing factors supress this shape change. In contrast to a control of motility by ZEB1, data-driven analysis of ESRP1/2 revealed that these splicing factors are largely negative regulators of cell elongation, but not motility. Elongated states, both motile (states L, D, B, I) and non-motile (states E, M), were upregulated upon ESRP1/2 depletion, but motile non-elongated states (state A) were not (Fig. 5g). This resulted in induction of the moderately locally invasive trajectory 7 upon ESRP1/2 depletion, but not highly motile trajectories 17 or 21 (Fig. 5i). Similar to ZEB1 depletion, ESRP1/2 knockdown also paradoxically increased the round, nonmotile trajectory 3 (though as expected in the opposite orientation to ZEB1 depletion) (Fig. 5h-j). This suggests that three general types of phenotype exist in this population: i) a mostly round and non-motile state, ii) phenotypes with varying shape states but without motility, and iii) highly motile states, irrespective of their shape. A ZEB1-ESRP1/2 pathway controls plasticity between these states. Accordingly, depletion of ZEB1 or ESRP1/2 reduced or enhanced invasion in orthogonal invasion assays (Supplementary Fig. 14c-e). These data confirm that Traject3d can be used in conjunction with bulk RNAseq of cell populations to identify key regulators of heterogeneity. Moreover, by analysing changes over time, Traject3d can be used to interrogate the specific cellular features that identified molecular regulators contribute to alternate phenotypes, such as cell shape by the splicing factors ESRP1/2 *versus* motility by ZEB1.

**An HGF ligand and c-Met receptor coupling control ZEB1 nuclear translocation**

Our data indicated that heterogeneity in expression of ZEB1-ESRP1/2 module underpins the appearance of invasive trajectories (trajectories 21, 7, 17) in Group II cells (GS689.Li, GS694.LAd, GS683.LALN, PC3-EMT, TEM4-18; Fig. 3b). Group III cells (JD1203.Lu, GS672.Ug, TEM2-5), however, express similar levels of ZEB1-ESRP1/2, yet lack trajectories 21, 7, 17 (Fig. 3b). This suggests that an additional factor is required to

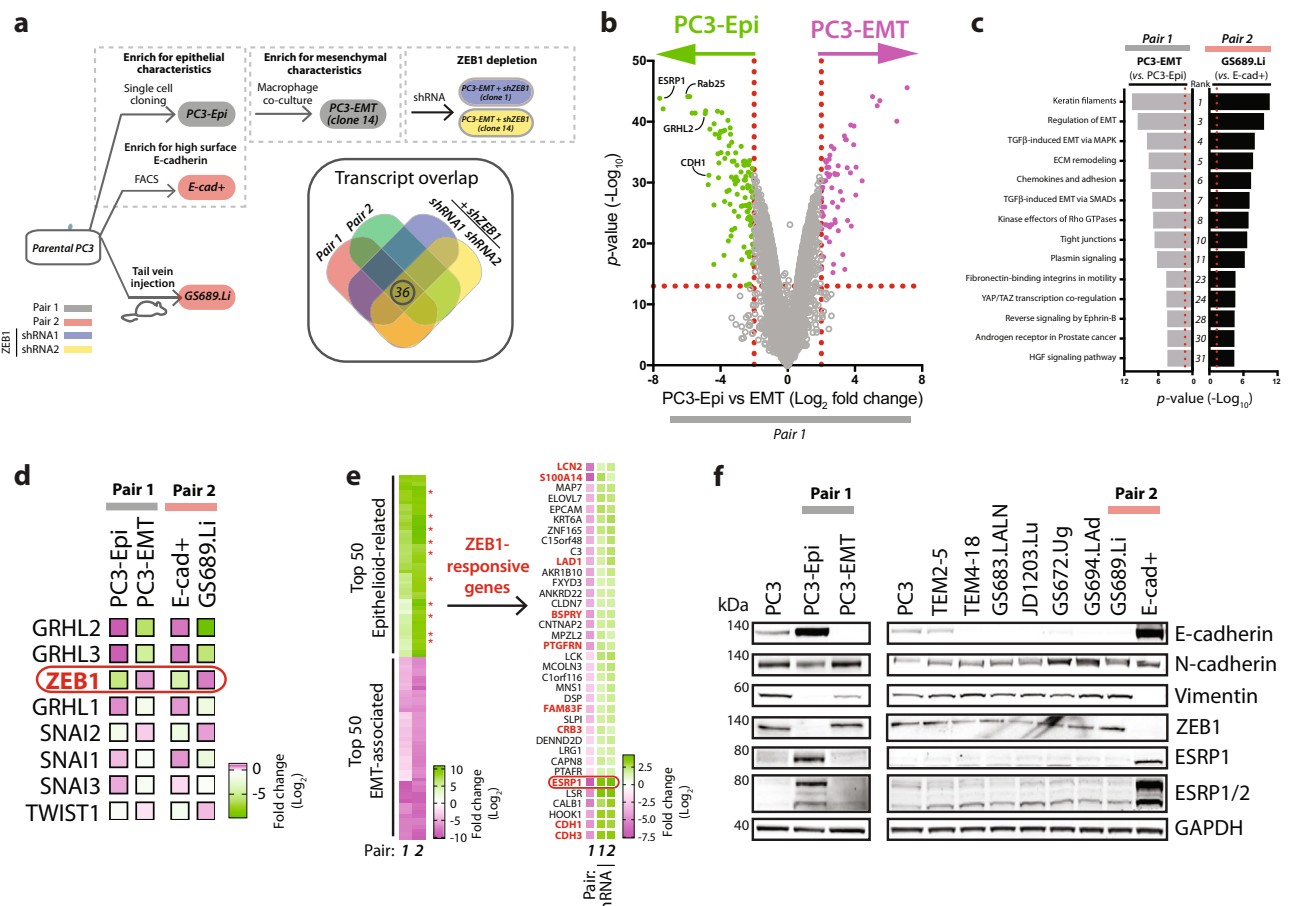

**Fig. 4 | Deconvolution of molecular pathways underpinning different phenotypes. a** Schema of PC3 subline pairs. Pair 1 is PC3 selected for epithelial shape (PC3-Epi) then made to undergo EMT by co-culture with macrophages (PC3-EMT). Pair 2 is PC3 FACS sorted for high surface E-cadherin (E-cad +) vs PC3 harvested from a liver metastasis following in vivo selection (GS689.Li). Two PC3-EMT lines stably expressing *ZEB1* shRNAs were also examined to identify ZEB1-influenced transcripts. Venn diagram summarizes RNAseq profiling which identifies 36 ZEB1-responsive genes upregulated in both Pair 1 and 2 and depleted upon *ZEB1* shRNA-induced knockdown. **b** Comparison of transcript levels (shown as Log$_2$ Fold Change) in Cell Pair 1 between epithelioid (PC3-Epi) vs invasive (PC3-EMT) samples. p-values shown as -Log$_{10}$ (Negative Binomial GLM fitting and Wald statistics using DESeq2). **c** MetaCore analysis of pathways maps from RNAseq profiling of PC3-Epi and PC3-EMT (Pair 1) and E-cad+ and GS689.Li (Pair 2) identified pathways commonly upregulated across both cell pairs in PC3-EMT (vs PC3-Epi) and in GS869.Li (vs E-Cad +). Molecular processes enriched in these data sets are ranked by *p* value (MetaCore version on 2017-10-26, -Log$_{10}$). **d** Comparison of expression of EMT transcription factors (TFs; GRHL1-3, ZEB1, SNAI1-3, TWIST1) between PC3-Epi and PC3-EMT (Pair 1) and E-cad+ and GS689.Li (Pair 2) cells indicates that ZEB1 is the most upregulated EMT TF in both cell pairs (in PC3-EMT and GS689.Li). Data are presented in the heatmap as Log$_2$ Fold Change from green to magenta. **e** Comparison of the top and bottom 50 most differentially expressed genes that were concordant between both cell pairs revealed that within the Top 50 epithelioid-associated genes, 20% of these transcripts were altered in expression upon shRNA targeting the transcriptional repressor ZEB1. ZEB1-responsive genes are indicated by red asterisk or red text. **f** Western blot analysis of PC3 sublines was performed using anti-E-cadherin, N-cadherin, vimentin, ZEB1, ESRP1, ESRP1/2 and GAPDH antibodies. GAPDH blot is loading control for vimentin and sample integrity control for other blots. Representative of *n* = 2 independent experiments. Note that ZEB1 had inverse expression compared to E-cadherin and ESRP1/2 in all PC3-derivative cell lines, except TEM2-5. Note that N-cadherin expression was not concordant with ZEB1.

drive these changes. Pathway analysis from RNAseq (Fig. 4c) indicated that differential expression of the HGF signalling pathway was associated with invasion, with HGF ligand but not c-Met receptor transcript levels induced in the invasive lines from Cell Pairs 1 and 2 (PC3-EMT, G5689.Li; Supplementary Fig. 15a). Addition of HGF to activate, or Cabozantinib to inhibit, c-Met signalling in parental PC3 cells robustly induced or completely abolished collective invasion, respectively (Supplementary Fig. 15b, c, d).

Application of broad user-defined classification suggested that HGF activation *versus* inhibition resulted in a switch between round *versus* spread and spindle shape (Supplementary Fig. 15e,f). This initially pointed to shape changes as a major function of HGF signalling. However, data-driven approaches, similarly to analysis of ZEB1, revealed that HGF signalling largely controls motility. Cabozantinib treatment led to a loss of motility associated with elongation (states L,

D, B, F, P, I, N; Supplementary Fig. 15g,h) and enrichment of non-motile trajectories (trajectories 20, 4, 5; Supplementary Fig. 15i). HGF stimulated the states with the highest motility (B, F, P, I, N; Supplementary Fig. 15h), but most prominently state A, which is a round state with extreme motility that is rare in parental PC3 cells. Accordingly, trajectory 17, to which state A is almost exclusively associated, was robustly induced upon HGF treatment (Supplementary Fig. 15i; green highlight box). This identifies HGF signalling as additionally required to ZEB1 to induce motile phenotypes.

We also examined whether HGF may act by controlling ZEB1 localisation. The identification of ZEB1 function in controlling motility rather than cell shape per se required live imaging, however, expression of exogenous, fluorescently tagged ZEB1 to track localisation would perturb cell behaviour. To overcome this, we examined endogenous ZEB1 localisation in fixed 3D spheroids classified into the three

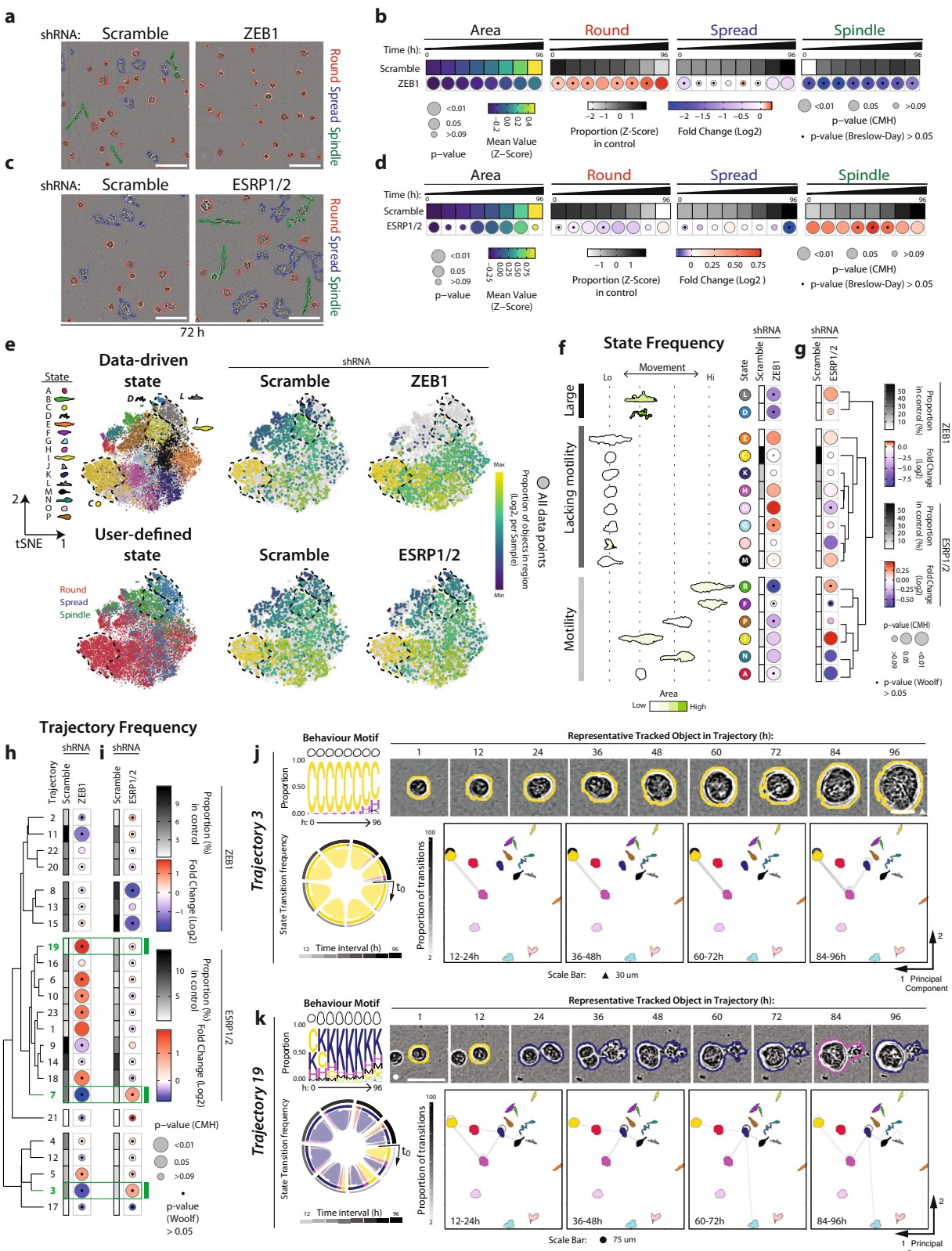

user-defined states (Round, Spread, Spindle) (Supplementary Fig. 16a), with and without HGF treatment. Total ZEB1 expression, as well as the ratio of nuclear:cytoplasmic localisation, was modestly higher in spheroids displaying Spindle shape in comparison to Round and Spread spheroids both at steady state and upon HGF treatment (Supplementary Fig. 16b, c). Notably, the addition of HGF only caused a modest alteration to ZEB1 total expression and a modest but

significant increase in ZEB1 nuclear translocation in Spindle-state spheroids. However, HGF treatment robustly increased the proportion of Spindle-state spheroids in the population (from 22.5% in DMSO condition to 49%) (Supplementary Fig. 16d). These data therefore suggest that the main effect of HGF is not to increase nuclear ZEB1 in spheroids already exhibiting Spindle state, but rather to increase the proportion of spheroids with Spindle features (which have the highest

**Fig. 5 | ZEB1 controls 3D motility features, not cell shape. a–d** Phase images and quantitation of PC3 spheroids expressing Scramble, (**a**, **b**) *ZEB1* or (**c**, **d**) *ESRP1/2* shRNA. Scale bars, 100 µm. Heatmaps show Area and Round, Spread or Spindle quantitation as described in Fig. 1h. *n* = 2 or 3 independent experiments respectively, 4 wells/condition/experiment, quantified in Supplementary Table 3. **e** t-SNE of spheroids from **a**–**d**. Plot points coloured by data-driven and user-defined state classifications. Purple-to-yellow shows per sample proportion of total objects in data-driven states, quantified before t-SNE. Black dashed lines, highlight regions corresponding to data-driven states. Data comprised of each spheroid identified in each image frame. t-SNE performed on 20,000 objects subsampled via GeoSketch, with iterations = 5,000,000, theta = 0.25, perplexity = 25. **f**, **g** Quantitation of data-driven state classifications. Representative outlines shown, arranged and coloured (light to dark green) by average class motility and area, respectively. Heatmaps show classification as Log$_2$ Fold Change from control (Scramble) (blue to red). Proportion of control in each class shown (white to black). Bubble size represents *p*-values (Cochran-Mantel-Haenszel test and Bonferroni adjusted), comparing classifications to control. Dots represent *p* value (Woolf test and Bonferroni-adjusted),

for homogeneity of odds ratio across experiments. Row order and dendrograms match Fig. 2e. n described in **a**. **h**, **i** Heatmaps show trajectory classification as Log$_2$ Fold Change from control (Scramble) (blue to red). Proportion of control is shown (white to black). p-values, Cochran-Mantel-Haenszel test to compare classifications to control, represented by bubble size. p-value, Woolf test for homogeneity of odds ratio across experimental replicates, represented by dots. *n* = 3 independent experiments, 3 wells/condition/experiment. Row order and dendrograms match Fig. 3b. n described in **a** and spheroids quantified in Supplementary Table 5. Trajectories discussed in text, highlighted in green. **j**, **k** Trajectory visualisation. Colours represent previously identified data-driven states and correspond to **e**–**g**. Behaviour motif depicting frequency of states in 12-h intervals, with outline of most abundant state shown. Phase images representing trajectory shown with outlines indicating state. Scale bars, 30 µm (triangle) and 75 µm (circle). Transitions shown as chord diagram, time interval greyscale. PCA used to arrange states in 2-dimensional space; transitions (proportion; greyscale) between (lines), and maintaining (circles), states overlaid.

---

ZEB1 nuclear levels of all three shapes). However, further investigation would be required to confirm a direct causal effect of increased nuclear ZEB1 on spindle shape and rule out the involvement of other mechanisms. Together, this emphasises the utility of combining Traject3d with existing bulk RNAseq or immunofluorescence approaches to identify molecular drivers of heterogeneity, such as the possible HGF-ZEB1-ESRP1/2 axis identified here.

### Traject3d identifies molecular combinations that can attenuate heterogeneity in treatment-resistant cells

We applied Traject3d to an independent set of treatment-resistant cancer cells to identify enrichment for specific states and trajectories, and to determine whether we could identify drug combinations that affect specific trajectories. We profiled additional PC3-derivative lines representing metastasis to liver (PC3M)[53] and resistance to Docetaxel (Dx) treatment (PC3M-D$^R$). We asked whether these subtypes would be dependent on the aforementioned HGF-ZEB1 pathway controlling plasticity and if inhibiting this pathway could restore treatment sensitivity (Fig. 6a).

ZEB1 was either expressed equivalently to (PC3M) or below (PC3M-D$^R$) parental PC3 levels, but PC3M and PC3M-D$^R$ displayed robust activation of c-Met, the latter of which could be effectively inhibited by the tyrosine kinase inhibitor Cabozantinib in all lines (Supplementary Fig. 17a). Both PC3M and PC3M-D$^R$ spheroids reliably displayed increased area from early timepoints throughout the time course (Supplementary Fig. 17b, c). Using broad, user-defined state classifiers, these spheroids exhibited increased Spread and Spindle states over time at the expense of roundness, with PC3M-D$^R$ particularly enriched for Spindle state. While Area was sensitive to Cabozantinib across cell lines, surprisingly, the use of user-defined classifications suggested that Cabozantinib only effectively inhibited Spread and Spindle state in the parental PC3 cells despite movies clearly showing decreased spindle and spread behaviours (Supplementary Fig. 17b, c). As expected, all phenotypes in PC3M-D$^R$ spheroids were refractory to Docetaxel treatment alone. The combination of Cabozantinib with Docetaxel treatment in PC3M-D$^R$ resulted in a delay to Spindle state induction (Supplementary Fig. 17c). Notably, Spindle shape did still occur at later timepoints but lacked a concurrent increase in Area and instead represented the shape of individual or dual cells rather than spheroids. The use of simple, user-defined classifications therefore gave the impression that PC3M and PC3M-D$^R$ are largely refractory to these treatments.

Traject3d performed superiorly in identifying how pharmacological perturbation of treatment resistance-associated phenotypes result from alternate cell states. Traject3d clarified that the apparent induction of both Spread and Spindle states in PC3M and PC3M-D$^R$ represented poor distinguishability between user-defined Spread and

Spindle classifications (L, I, N) (Fig. 6b-d, see Supplementary Fig. 17d for comparison). Traject3d allowed discrimination of large, motile, elongated states that were common to both PC3M and PC3M-D$^R$ spheroids (D, I, L) from those that were specific to PC3M-D$^R$ spheroids (N) (Fig. 6d). Moreover, Traject3d distinguished states with equivalent user-defined classifications that were differentially sensitive (L, D) or refractory (I, K, N) to Cabozantinib alone, as well as those specifically inhibited by dual Cabozantinib and Docetaxel treatment in PC3M-D$^R$ spheroids (A, I, N). Notably, two rare states (A, N) were induced specifically in PC3M-D$^R$ spheroids and could only be inhibited by the combination of Cabozantinib and Docetaxel treatment. Rare cell state A is notable in that it overlapped with Round classification but displayed higher motility than other 'round' states. This highlights the utility of Traject3d in identifying treatment resistance-associated behaviour states, as seemingly similar user-defined states can represent distinct biological phenotypes with differential sensitivity to interventions.

Reconstructing state changes over time provided clarity as to how alternate states can give rise to alternate phenotypes, particularly in drug-sensitive *versus* drug-insensitive scenarios. This revealed multiple types of behaviours that had been induced in parallel in PC3M and PC3M-D$^R$. This included trajectory 16, typified by elongated but poorly motile states K and M (Fig. 6d-f). It also included three distinct types of motility phenotypes. Both cell types induced the modestly invasive trajectory 7 (Fig. 3d, Supplementary Figs. 11-12) typified by state I (Fig. 6e). PC3M displayed a second type of motility: induction of the very large, motile, elongated trajectory 21 (Fig. 3e, Supplementary Figs. 11-12) typified by the large, modest motility state L (Fig. 6e). PC3M-D$^R$ possessed equivalent levels of both above invasive phenotypes but added a third type: trajectory 17 (Fig. 6g, Supplementary Figs. 11-12), comprised of extremely motile spheroids typified by the round, highly motile state A (Fig. 6e). Notably, each of these three motile trajectories was differentially sensitive to treatment. Trajectory 17 was sensitive, while trajectories 16 and 7 were refractory, to Cabozantinib treatment; only dual Cabozantinib/Docetaxel treatment was successfully able to abolish trajectory 17 (Fig. 6e). This emphasises that induction of biological features, such as metastasis or drug-resistance, can be associated with multiple, parallel phenotypes being induced, such as the three distinct types of invasion in PC3M-D$^R$. Given that each of these invasion states is differentially drug sensitive in the same sample – something that was not detected by simple user-defined classifications – this emphasises the importance of data-driven heterogeneous phenotype detection over time when profiling 3D cultures that Traject3d now provides.

### Application of Traject3d to organoids grown in dome culture
As a complement to the 3D culture and imaging approach utilised above we applied Traject3d to an alternate modality: the imaging of

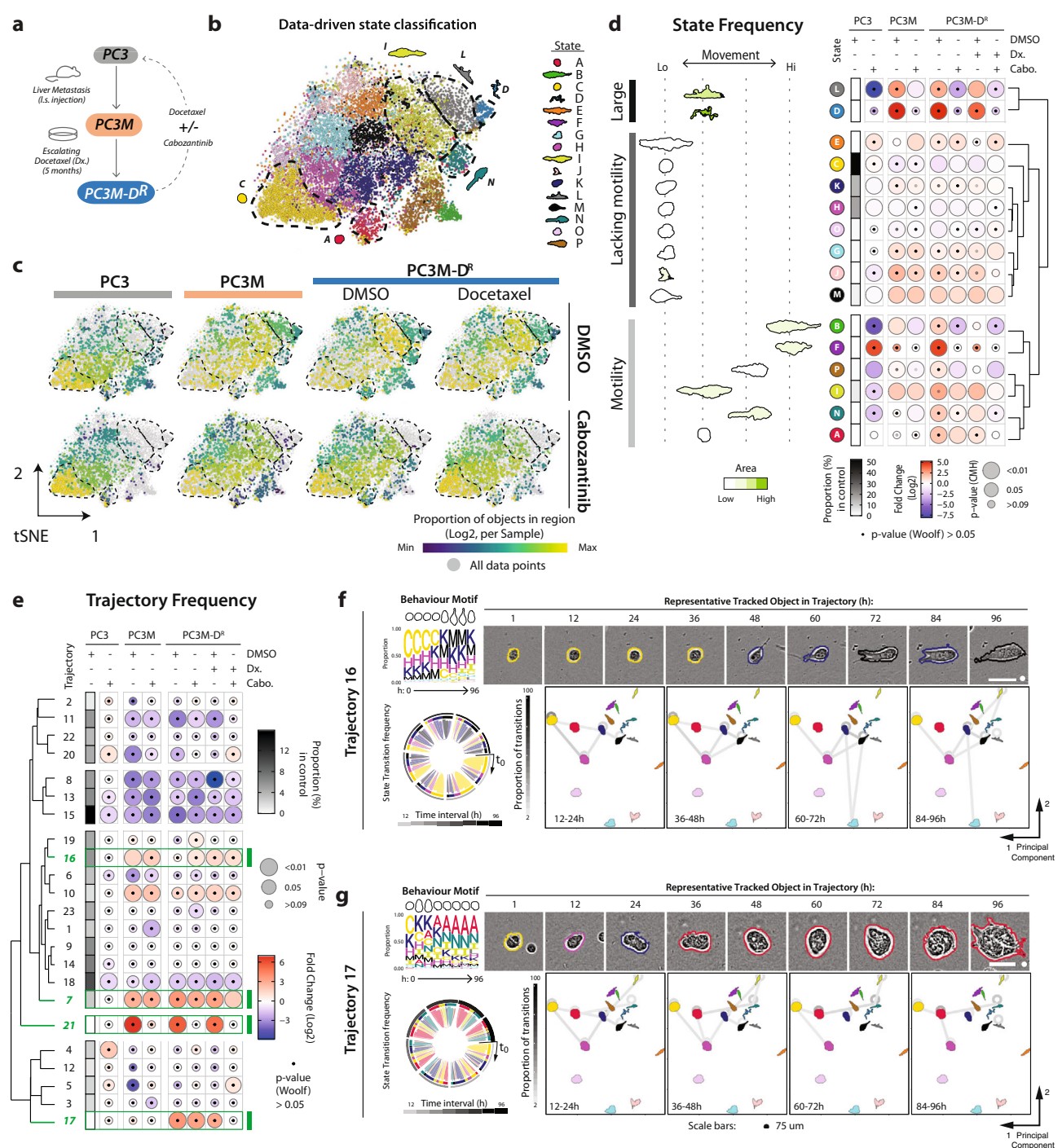

**a**

**b** Data-driven state classification

**c** PC3 | PC3M | PC3M-D^R

**d** State Frequency

**e** Trajectory Frequency

**f** Behaviour Motif / Representative Tracked Object in Trajectory (h): — Trajectory 16

**g** Behaviour Motif / Representative Tracked Object in Trajectory (h): — Trajectory 17

Scale bars: ■ 75 um

organoids grown in dome-based culture (Supplementary Fig. 18a). We examined primary organoids from a genetically engineered mouse model of colorectal cancer, intestine-specific mutant K-ras, loss of TP53 and activation of Notch signalling (*villin*Cre^ER *Kras*$_{G12D/+}$ *TrpS3*$^{fl/fl}$ *Rosa26*$^{Nlicd/+}$; KPN)[54]. We included stimulation with two EMT-associated ligands, TGFβ1 and IL-6, to examine potential effects on organoid behaviour. We imaged 97,697 objects and tracked 11,295 object series over multiple days (Supplementary Table 6). Imaging every two hours for 6 days indicated that cells formed stereotypical spherical organoids varying in size by 6 days, with lumens already apparent in many organoids by two days (Supplementary Fig. 18b). Data-driven identification revealed 14 cell states (A-N; Supplementary Fig. 18c). This included the prototypical organoid morphology of a monolayer of cells around a single central lumen (states H, L). Also detected were

instances of clusters of cells not (at that point) forming into prototypical organoid morphology (e.g. states A, D, E, G). As a quality control, instances where air bubbles were detected as objects (state I) were separated on the t-SNE maps from other objects, as were instances of organoids merging (state C) (Supplementary Fig. 18c), both of which were extremely rare (see proportion of events in Control in Supplementary Fig. 18d).

Examination of cell state alone revealed each treatment (TGFβ1, IL-6, or vehicle control) could increase the frequency of large motile states (states J, K) that were very rare in control (0.5% and 0.2%, respectively) but still only resulting in rare occurrences (0.8-1.3% and 0.4-1.2%, respectively). TGFβ1 alone modestly decreased the frequency of prototypical spheroid organoids (states H, L; Supplementary Fig. 18d). Importantly, when restoring time to tracked objects to

**Fig. 6 | Data-driven analysis identifies distinct phenotypes with differential sensitivity to pharmacological treatment. a** Schema, subline derivation from liver metastases in nude mouse bearing splenic explant of PC3. PC3M cells were treated with escalating doses of Docetaxel (Dx.) until resistant (PC3M-D$^R$) then treated with Cabozantinib. **b, c** t-SNE of spheroids treated with Cabozantinib and/or Docetaxel. Plot points in **b** coloured by data-driven state classifications. Purple-to-yellow in (**c**) shows per sample proportion of total objects in each data-driven state, quantified prior to t-SNE. Black dashed lines, highlights regions corresponding to data-driven states. Data comprised of each spheroid identified in each image frame. $n = 3$ independent experiments, 4 wells/condition/experiment. Spheroids quantified in Supplementary Table 3. Analysis performed on 20,000 objects subsampled via GeoSketch, with iterations = 5000, theta = 0.25, perplexity = 100. **d** Quantitation of data-driven state classifications. Representative outlines shown, arranged and coloured (light to dark green) by average class motility and size, respectively. Heatmap shows classification as Log$_2$ Fold Change from control (PC3 + DMSO) (blue to red). Proportion of control in each class shown (white to black). Bubble size represents $p$ values, Cochran-Mantel-Haenszel test (Bonferroni adjusted), to compare classifications to control. Black dot represents p-value, Woolf test (Bonferroni-adjusted), testing homogeneity of odds ratio across experiments. Row order and dendrograms match Fig. 2e. n described in **b**. **e** Heatmap shows trajectory classification as a Log$_2$ Fold Change from control (PC3 + DMSO) (blue to red). Proportion of control is shown (white to black). Bubble size represents p-values, Cochran-Mantel-Haenszel, comparing classifications to control. Black dot represents $p$ value, Woolf test, testing homogeneity of odds ratio across experiments. Row order and dendrograms match Fig. 3b. Spheroids quantified in Supplementary Table 5. Trajectories discussed in text, highlighted in green. **f, g** Trajectory visualisation. Colours represent previously identified states and correspond to **b, d**. Behaviour motif depicting frequency of states in 12-hour intervals, with outline of most abundant state shown. Phase images representing trajectory shown with outlines indicating state. Scale bars, 75μm (round (**f**)) and 30μm (triangle (**g**)). Transitions shown as chord diagram, time interval greyscale. PCA used to arrange states in 2-dimensional space; transitions (proportion; greyscale) between (lines), and maintaining (circles), states overlaid.

identify and quantify trajectories, we identified that despite modest effects on some cell states (Supplementary Fig. 18d) there was no difference in phenotype of these organoids over time with any of the applied treatments (TGFβ1, IL-6, or vehicle control), other than extremely minor changes in the presence of rare air bubbles in the culture. It is notable that while the most frequent trajectory observed captures a stereotypical presentation of organoid formation (trajectory 6), there are not infrequent co-occurring phenotypes, including where clusters of cells do not give rise to an organoid (trajectories 1, 3, 4), and clusters of cell aggregates without (trajectory 7) or with (trajectory 5) merging with other objects (Supplementary Fig. 18e,f). This emphasises how multiple, often unreported, phenotypes can co-occur within a sample. Traject3d provides a methodology to now detect such populations, enabling future testing of their biological significance. Moreover, this emphasises that the addition of time from live imaging is essential to detect the contribution of alternate cell states to phenotype.

## Discussion

We here described Traject3d, a method for quantitative, data-driven identification of distinct states over time from label-free multi-day time-lapse imaging of 3D culture. This allows identification and molecular dissection of a range of cellular states based on morphological data exhibited by cells as they expand in culture to form 3D multicellular structures. The identified trajectories are intrinsically distinct on a biological level, illustrated here by differential metastatic/invasive potential and drug resistance. This approach is equally applicable to other signalling and phenotypic distinctions for any 3D culture of interest.

Traject3d allowed us to answer simple, yet fundamental, questions regarding 3D culture. First, it provides quantitative demonstration that heterogeneity in 3D culture is widespread across a variety of cell types. Second, it reveals that heterogeneity represents the presence of multiple independent phenotypes that occur in parallel. These are biologically distinct, such as multiple modes of invasion occurring in the same sample in parallel yet being differentially sensitive to pharmacological or genetic perturbation. This contrasts to the analysis of single-cell omics technologies in which a limited number of terminal snapshots of state are computationally inferred into a predicted sequence, usually by detecting bifurcation points when different states are detected[8–14]. Traject3d uses real-time tracking to identify the true state order. This reveals that alternate phenotypes can result from exhibiting a unique combination of states (bifurcation points), but can also occur by using the same states in a different order. Moreover, several phenotypes are distinguished by motility features, which are unlikely to be identified from a low number of terminal snapshots of state. Therefore, by identifying how cell states can lead to

alternate phenotypes co-occurring within the same cell sample, Traject3d is a new complementary technology for the expanding arsenal of single cell technologies.

One powerful feature of Traject3d is the ability to deconvolve bulk RNAseq to identify biologically important pathways controlling alternate phenotypes. As proof of application, we identified that HGF signalling is one pathway controlling an alternate phenotype. The granular analysis that Traject3d provides distinguished that while an elongated state is usually associated with invasion, rather than controlling shape per se, ZEB1 controls cell motility. Importantly, invasion can occur in small, round, highly motile cells. By implication, inference of invasive potential solely from shape[1–7] will likely under-power prediction of metastatic capabilities as instances of both elongated, non-motile states and instances of non-elongated, highly motile states can and do occur. The biological importance of this is underscored by our finding that three differentially drug-responsive motility phenotypes co-occur in parallel in drug-resistant PC3M-D$^R$ cells, including a non-elongated shape, yet highly motile phenotype. Notably, Met levels and signalling are upregulated in advanced prostate cancer patients, associated with Olaparib-resistance in prostate cells and regulate invasion in vitro and tumourigenesis in vivo[55], suggesting that such phenotype plasticity might be a general feature of drug-resistant cells.

In our experiments, distinct states also occurred near each other, suggesting either cell-autonomous behaviours or alternate capacity of neighbouring cells to respond to environmental cues. Some states were present in all trajectories, some were enriched across fewer trajectories, and others were largely but non-exclusively restricted to specific trajectories. Most trajectories, however, even those with distinct morphogenesis patterns, share a similar repertoire of states. This indicates that distinct phenotypes can occur by using rarer states at differing frequencies or in a different temporal order. An essential advancement that Traject3d provides is to therefore untangle these possibilities by analysing cells states in true temporal order from live imaging.

We explored two approaches for defining states, user-defined and data-driven de novo identification. Both approaches can identify different states present in 3D culture, even at low frequencies. Although user-defined classifications enable interrogation of specific phenotypes of interest, the data-driven approach has the advantage of identifying subtle differences in phenotype that are beyond user inference, such as the incorporation of motility features that were essential for the definition of several phenotypes. A caveat for data-driven approaches in general is the requirement for collection of all data for processing together *en masse* to determine common and distinct cell states. This may not be practical for all applications. Therefore, a user-defined classification may be preferred during data collection and project development before final analysis with granular

clarification of phenotypes. This makes these two methods valuable companion approaches.

Given the complexity of live imaging, an open question is if this is needed *versus* simple evaluation of limited timepoints. Our analyses show that this is not a one size fits all answer; in PC3-derivative lines, alternate phenotypes occur through induction of alternate rarer cell states or by using similar cell states in different ways. While, with enough static samples, one may capture enough heterogeneity to detect the former route for alternate phenotypes, this approach would not detect the latter. In contrast, in examining tumour-derived organoids, although some rarer states could be induced with select treatments, the overall phenotypes were largely unaffected across time. This suggests that inferring complex phenotypes from limited timepoints likely underappreciates complex phenotypes that live imaging can reveal, and in the case of organoids, could aberrantly over-extrapolate rare states as indicating different ending phenotypes.

In this work we used Traject3d to identify the landscape of temporal states in a series of cancer cell lines with varying metastatic capacities and differential drug resistance. While we can detect heterogeneity in 3D with this method, we cannot a priori predict the behaviours that will occur in a tumour in vivo. Moreover, we can only sample in vitro that which is isolated from a patient during sample derivation. Consequently, our method does not indicate all heterogeneity patterns that can and do exist in vivo but gives insight into which parallel phenotypes occur in materials grown ex vivo. Further validation of these phenotypes in vivo is required for full elucidation of the impact of heterogeneity within patient tumours.

Parallelisation of live imaging was essential for identifying distinct cell states with robust statistical support, which we achieved by adapting 3D culture approaches to work with perfect-focus algorithms in 96-well plates using the IncuCyte system. In principle, Traject3d could be applied to the analysis of imaging data from other systems, provided that the resulting images can be segmented, with analysis results depending heavily on the quality of image segmentation and object tracking. Due to the relatively slower growth and movement dynamics of 3D culture compared to 2D approaches, inbuilt approaches in CellProfiler are sufficient for analysis of many 3D cultures but may need to be adapted for some downstream uses. CellProfiler pipelines that successfully work for segmentation and tracking across multiple 3D cultures are provided in the Traject3d GitHub repository. It may also be possible to use other image segmentation methods, such as ilastik[56] or machine learning/neural networks that may be better at distinguishing objects that are touching or images with low contrast between objects and background. Images segmented in this way can then be loaded into CellProfiler to generate the measurements in a format compatible with use by Traject3d.

A larger consideration is that no amount of improved tracking or segmentation will compensate for the fact that in 3D objects move and can consequently collide, merge, and split. This isn't simply poor segmentation or tracking, but rather bona fide events and a reality for the experimentalist. To identify distinct recurring morphogenesis patterns (trajectories) we need to compare tracked objects over the same period of time. We did this for 18,922 spheroids objects and 11,295 objects from organoid studies over multiple days. Despite this extremely large number of tracked objects analysed this meant that we are unable, by definition, to include objects tracked for shorter periods of time. This could occur for many reasons, including splitting/merging, but also because an object touches the edge of, or leaves, the field of view. Therefore, some data needs to be filtered out of imaging so that the data that makes it through this quality control is robust and appropriate. The inherent limitation to our approach, as with many imaging approaches, is therefore that Traject3d analysis can only detect those events that can be accurately segmented and tracked. Despite detecting complex, varied 3D behaviours this may possibly underestimate even stronger heterogeneity due to objects that we have been unable to track extensively. As a design principle, we do not exclude such merging/splitting events based on a priori assumptions of their value or lack thereof. Rather, we include capture of these events where possible, and allow the user to conclude biological importance (such as presentation of when air bubbles or merging events occur).

In this work we analysed 3D cell line spheroids grown in a singular plane for ease of imaging, as well as tumour-derived primary organoid cultures grown in domes. Both approaches were equally amenable to Traject3d, though the method utilised in PC3 cultures results in minimal ECM use and cost and allows capture of more objects per field of view. Traject3d could therefore be applied to other live-imaged 3D culture modalities, provided accurate segmentation and tracking can be performed on sufficient sample sizes. Moreover, Traject3d could be expanded to include signalling reporters from fluorescence imaging for distinct cell states or used to investigate the effect of changing the microenvironment through altering extracellular matrix embedding of cells. Similarly, Traject3d could also be adapted to study co-culture using fluorescently tagged labels or dyes to label different cell types. The challenge in this would be determining what happens computationally when these populations physically interact (i.e. come into contact), as currently the design of Traject3d is based on what happens to distinct objects in parallel. In addition, although we focus on tumour cell lines and organoids as proof of principle, Traject3d could be equally applied to alternate systems such as induction of pluripotent stem cells into distinct cell types, the differential sensitivity to 3D cultures to treatment with pharmacological agents or immune cells, or any other system that utilises 3D culture. Moreover, combining Traject3d's ability to detect heterogeneous parallel phenotypes with multiplex barcoding, CRISPR and cell isolation technologies may unlock the potential to perform functional genomics in 3D, by allowing direct observation of genotype-to-phenotype manipulations in large-scale screens.

A crucial question is how the identification of distinct phenotypes within a 3D culture that Traject3d provides aids the experimentalist? We propose that by identifying co-occurring phenotypes in a sample, Traject3d allows interrogation of an overlooked phenomenon in biology: an incomplete response. For example, if two populations co-exist in a sample, one that is resistant to perturbation while the other is sensitive, the overall response of the sample depends on the ratio of these two subpopulations. In perturbation of a sample, a modest response - as determined by the average response of all cells – could in reality be a complete response in one subpopulation concurrently with insensitivity in the other. Conversely, a robust – yet incomplete – response may relate to sensitivity of the majority of the cells, and insensitivity of a numerically minor but phenotypically important subpopulation of cells. An incomplete response, and the assumption of acquired resistance, is a key clinical problem in oncology. Without tools to identify heterogenous, parallel subpopulations, whether de novo resistance is acquired upon drug treatment, or whether a rare and already drug-resistant subpopulation existed that becomes expanded upon treatment is unclear. As one potential application of heterogeneity analysis, Traject3d provides the tools to distinguish these possibilities, and opens the door to identify perturbation combinations that affect all subpopulations, perhaps resulting in a complete response across all subpopulations.

With the increasing utilisation of 3D approaches to understand cell behaviour, not only in cancer biology but also developmental biology, regenerative medicine, neuroscience, and many other fields, analysis of the associated complexity of 3D phenotype(s) and how they vary over time has presented a major challenge. We therefore present Traject3d as an open-source software pipeline to identify spatio-temporal multicellular heterogeneity in 3D culture.

## Methods

### Cell Culture

Parental PC3 (ATCC, CRL-1435) and PC3 variants: PC3 E-cad + , TEM4-18, TEM2-5, GS689.Li, GS694.LAd, GS683.LALN, JD1203.Lu, GS672.Ug (M. Henry, University of Iowa), PC3-Epi and PC3-EMT (K. Pienta, Johns Hopkins School of Medicine), PC3M, PC3M DR and CWR (H. Leung, Beatson Institute) cells were maintained in RPMI-1640 (Gibco), 10% fetal bovine serum (FBS; Gibco) and 6mM L-glutamine (Gibco). PC3M-DR cells were cultured in 2 nM Docetaxel (Sigma). RWPE-1 and RWPE-2 cell lines (ATCC, CRL-11609 and CRL-11610) were maintained in keratinocyte serum-free medium (K-SFM) supplemented with 50 µg/ml BPE and 5 ng/ml EGF (all Gibco). 22Rv1 cells (H. Leung, Beatson Institute) were grown in phenol-free RPMI-1640 (Gibco), 10% charcoal-stripped FBS and 6mM L-glutamine. MDCK-II (K. Mostov, UCSF) and Caco-2 cells (L. Machesky, Beatson Institute) were cultured in minimum essential medium (MEM; Gibco) containing 5% or 20% FBS, respectively. MDA-MB-231 (M. Olson, Beatson Institute), $Pdx1$-Cre, LSL-$Kras^{G12D/+}$ (KC), $Pdx1$-Cre, LSL-$Kras^{G12D/+}$ LSL-$Trp53^{R172H/+}$ (KPC), $Pdx1$-Cre, LSL-$Kras^{G12D/+}$ LSL-$Trp53^{fl/+}$ (KPflC) and $Pdx1$-Cre, LSL-$Kras^{G12D/+}$, $Pten^{fl/+}$ cells (Pten) (J. Morton, Beatson Institute) were grown in Dulbecco's Modified Eagle's Medium (DMEM; Gibco), 10% FBS and 6mM L-glutamine. 293-FT (Thermo Fisher Scientific) were cultured in DMEM, 10% FBS, 6mM L-glutamine and 0.1 mM non-essential amino acids (NEAA).

Growth factors or inhibitors were added as follows: 50 ng/ml Hepatocyte Growth Factor (HGF) (PeproTech) and 10 µM Cabozantinib (Bio-Techne). Cells were screened for mycoplasma contamination routinely. PC3, RWPE-1 and RWPE-2 cells were authenticated using short tandem repeat (STR) profiling.

### Generation of stable cell lines

Stable knockdown was achieved by co-transfecting HEK 293-FT packaging cells with pLKO.1 shRNA plasmids with lentiviral packaging vectors (VSVG and SPAX2, Addgene) using Lipofectamine 2000 (Thermo Fisher Scientific). Viral supernatants were collected, cell debris removed using PES 0.45µm syringe filters and then concentrated, as per the manufacturer's instructions, using Lenti-X Concentrator (Clontech). PC3 cells were then transduced with the lentivirus for 3 days before selection with 2.5 µg/ml puromycin (Thermo Fisher Scientific). Sequences of shRNAs used are listed in Supplementary Table 7.

### RNAseq data analysis

mRNA expression data were mined from publicly available RNAseq data sets for PC3 sublines: PC3 E-cad+ vs GS689.Li; SRS354082, or PC3-Epi versus PC3-EMT14; GSE4823[30,31].

Data were examined for differential expression analysis using the R environment version 3.4.2, utilizing packages from the Bioconductor data analysis suite[57]. Differential gene expression was analysed based on the negative binomial distribution using the DESeq2[58] package version 1.18.1.Pathway Maps analysis was performed using MetaCore from Clarivate Analytics (https://portal.genego.com; November 2017). Plots were generated using Prism 8 (GraphPad).

### Immunoblotting

$2 \times 10^5$ cells were plated per well on a 6-well plate for 24 hours then treated with the growth factors or inhibitors described above for a further 24 hours. Plates were washed once with ice cold PBS then RIPA buffer was added for 15 minutes (50 mM Tris-HCl, pH 7.4, 150 mM NaCl, 0.5 mM MgCl$_2$, 0.2 mM EGTA, and 1% Triton X-100 with cOmplete protease inhibitor cocktail and PhosSTOP tablets (Roche)). Cells were scraped and samples clarified by centrifugation at 15,800 g at 4 °C for 15 minutes. BCA Protein Assay kit (Pierce) was used as per the manufacturer's instructions to determine protein concentration. SDS-PAGE was then performed on 4-12% gradient gels using 20 µg lysate and proteins transferred to PVDF membranes using the iBlot 2 transfer system (Thermo Fisher Scientific). Membranes were incubated in Rockland blocking buffer (Rockland) for 1 hour with agitation and primary antibodies added overnight at 4 °C (all 1:1,000, unless stated otherwise). Antibodies used were as follows: anti-GAPDH (14C10) (CST 2118 (1:5,000)), anti-Met (25H2) (CST 3127), anti-Met phospho 1234/1235 (D26) (CST 3077), anti-N-cadherin (D4R1H) (CST 13116), anti-vimentin (V9) (Santa Cruz, sc-6260), anti-E-cadherin (Clone 36) (BD Biosciences 610181), anti-ESRP1 (210-301-B89S) and anti-ESRP1/2 (210-301-C31S) (both Tebu-bio) and anti-ZEB1 (Sigma HPA027524). Membranes were washed with 1x TBST three times for 10 minutes and appropriate secondary antibodies added for 1 hour. After three 15-minute washes in 1x TBST membranes were imaged using a ChemiDoc Imager (BioRad) after exposure to ECL Extreme or ECL Pico (Expedeon) or using an Odyssey Imaging System (LI-COR Biosciences). Bands were quantified using either Image Lab 6.1 (BioRad) or Image Studio Software 6.0 (LICOR Biosciences). Number of independent experiments ($n$) is stated in the appropriate Figure Legend and quantitation is shown as mean ± s.d. p values are shown on each graph as follows; n.s. = not significant, $*p \leq 0.05$, $**p \leq 0.01$, $***p \leq 0.001$ and $****p < 0.0001$. GAPDH was used as a loading control for each immunoblot and a representative image for each sample set is shown where appropriate. anti-ZEB1 (Sigma HPA027524), anti-ESRP1 (210-301-B89S), anti-ESRP1/2 (210-301-C31S), anti-N-cadherin (D4R1H), anti-Met (25H2) (CST 3127) and anti-Met phospho 1234/1235 (D26) (CST 3077) were validated by western blot using specific shRNAs or inhibitors. Additional validation information is available from manufacturers.

### Invasion assay

96-well ImageLock plates (Essen Biosciences) were coated with 10% Growth Factor Reduced Matrigel (GFRM) (BD Biosciences) diluted in medium overnight at 37 °C. 70,000 PC3 cells were re-suspended in 100 µl medium per well and plated for 4 hours at 37 °C. The resultant monolayer was wounded using a wound-making tool (Essen Biosciences), washed twice with medium and overlaid with 25% GFRM (50 µl) for an hour. 100 µl medium was then added and plates imaged every hour for up to 4 days using the IncuCyte® ZOOM or S3 (Essen Biosciences). Growth factors or inhibitors were added, at the concentrations described above, to both the GFRM and to the medium. Incucyte ZOOM Live Cell Analysis System Software 2018A or Incucyte S3 Live Cell Analysis System (2021 A) (Essen Biosciences) were used to calculate Relative Wound Density (RWD) for each well. Results are presented as RWD for the timepoint at which the average RWD of the control samples is 50% ($T_{max}1/2$). Number (n) of independent experiments and replicates/condition are detailed in appropriate Figure Legends. Values, mean ± s.d. $p$ values were calculated using Students t-test (two-tailed) or One-Way ANOVA and are shown on each graph as follows n.s. = not significant, $*p \leq 0.05$, $**p \leq 0.01$, $***p \leq 0.001$ and $****p < 0.0001$.

### Fixed 3D spheroid culture and analysis

Single cell suspensions were made ($1.5 \times 10^4$ cells per ml) in the appropriate medium supplemented with 2% GFRM. 150 µl of this mix was plated per well in a 96 well plate (Greiner) precoated with 10 µl of GFRM for 15 minutes at 37 °C. DMSO or 50 ng/ml HGF was added for 24 h after 2 days. Wells were then washed with PBS and fixed in 4% paraformaldehyde for 15 minutes. Samples were blocked in PFS (0.7% fish skin gelatin/0.025% saponin/PBS) for 1 hour at room temperature (RT) with gentle shaking and anti-ZEB1 antibody (Sigma, HPA027524) added overnight (1:100) at 4 °C. After three washes in PFS, Alexa Fluor 488 Donkey anti-rabbit secondary antibody (1:200, A21206), HCS CellMask™ Deep Red Stain (1:50000, H32721) and Hoechst 34580 (1:1000, H21486) (all Thermo Fisher Scientific) were added for 45 min at RT. Samples were maintained in PBS, after 3 × 5-min washes in PBS, until imaging was carried out. Spheroids were imaged using an Opera

Phenix™ High Content analysis system (x64 objective). Each spheroid was imaged in 30 consecutive planes (2μm step size).

Harmony High-Content Imaging and Analysis Software (PerkinElmer, Version 4.6) was used to analyse maximum intensity projections of all planes. For 3D morphology assay Hoechst and HCS CellMask were used as described above to define nuclei and whole spheroid shape, respectively. Used in combination these dyes allowed the generation of a nuclear mask (all nuclei) and a cytoplasmic mask (whole spheroids excluding all nuclei) for each spheroid. Total intensity of anti-ZEB1 staining (with Alexa Fluor 488 Donkey anti-rabbit secondary antibody) was then measured in the nuclei (N) and in the cytoplasmic region (C) of each spheroid. Spheroids in contact with the image border were excluded. The morphological properties of each spheroid were calculated to classify them into three subpopulations (Round, Spread and Spindle) using machine learning following manual training. Data is presented in box and whiskers plots as nuclear to cytoplasmic ratio (N:C) of total ZEB1 intensity or total ZEB1 intensity for each spheroid/subpopulation/treatment. The proportion of each spheroid subpopulation (Round, Spread, Spindle) is also presented for each treatment. $n = 2$ experimental replicates with 4 technical replicates/condition. P-values (Student's 2-tailed t-Test): ***$p \leq 0.001$ and ****$p \leq 0.0001$. 965 (DMSO) and 875 (HGF) spheroids were imaged in total.

Single-cell suspensions of PC3 were also set up as above, fixed and stained with Hoechst (1:1000) and Alexa Fluro 568 Phalloidin (Thermo Fisher, A12380) (1:200). Spheroids were imaged using an Opera Phenix™ High Content analysis system (x20 objective). Each spheroid was imaged in 43 consecutive planes (2μm step size). $n = 3$ experimental replicates with 3 technical replicates/condition.

## 3D spheroid culture and live imaging
Single cell suspensions were made as described above in 96 well ImageLock plates (Essen Biosciences). HGF, Cabozantinib or Docetaxel were added at concentrations described above. Plates were then incubated at 37 °C for 4 h then imaged using an IncuCyte® ZOOM or S3 (Essen Biosciences). Images were taken every hour for 4−6 days at 2 positions in the middle of each well using a 10x objective lens. Number of independent experiments (n), technical replicates/condition and number of spheroids quantified are stated for each experiment in corresponding Figure Legend and in Supplementary Tables 2, 3, and 5.

## Organoid culture and live imaging
The murine *villin*[CreER]; *Kras*[G12D/+]; *Trp53*[fl/fl]; *Rosa26*[NIicd/+] organoid line RBVKPN RKAC13.1 g, generated in[54], was cultured in Advanced DMEM/F12 (ADF, Gibco) supplemented with 2mM L-Glutamine, 10 mM HEPES, 100U/ml penicillin-streptomycin, 1x B27-supplement and 1x N2-supplement (all Gibco), adjusted to complete culture medium (CCM) with 50 ng/ml Recombinant Human EGF and 100 ng/ml Recombinant Murine Noggin (both Peprotech). Organoids were maintained in domes of 20 μl GFRM (BD Biosciences) in 6 well plates containing 2 mL of CCM at 37 °C, 5% $CO^2$.

Organoids were washed in cold PBS to remove GFRM and dissociated into small clusters and single cells using StemPro Accutase (Gibco) at 37 °C for 10 min. Dissociation was stopped with equal volume of sterile 1% BSA (Merck) in PBS and cells centrifuged at 600 $g$ for 5 min at 4 °C. The supernatant was removed and cells re-suspended in 1 ml ADF medium. Cells were combined with an equal volume of Trypan Blue (Gibco), counted and adjusted to $2 \times 10^4$ cells/ml in GFRM. Whilst kept on ice, 10 μl GFRM domes were plated in the centre of 96-well flat bottom TC-treated microplate (Corning). Plates were tipped upside down to allow domes to hang whilst polymerizing at 37 °C for 10 min. CCM (100 μl) containing either vehicle (0.1% BSA/PBS), 5 ng/ml TGFβ1 or 10 ng/ml IL-6 (both Bio-Techne) was placed over each dome and the plate equilibrated at 37 °C for 30 min.

Plates were then imaged using the IncuCyte® S3 IncuCyte Organoid Analysis Software Module. Entire wells were imaged every 2 hours for 6 days using a 4x objective lens. Cultures were maintained by replenishing CCM every 3 days. This module was also used to generate outlines of imaged objects, which were then exported into CellProfiler for analysis.

## Statistics & reproducibility
No statistical method was used to predetermine sample size and no data were excluded from the analyses. The experiments were not randomized and the Investigators were not blinded to allocation during experiments and outcome assessment. P values and the specific statistical test and adjustments used for each experiment are described in relevant Figure Legends and Methods Section.

## Generating rules for user defined classification of 3D PC3 spheroids
We used CellProfiler[18] (https://cellprofiler.org/, v3.1.8) to process the phase images acquired using the IncuCyte system, tracking spheroids through the time course of the experiment, and generating a database which contained measurements of size, shape, and movement features. This CellProfiler pipeline consisted of the following modules and functions: *Images* (image input), *Metadata* (metadata extraction), *NamesAndTypes* (assignment of meaningful names), *Groups* (grouped data sets), *ColorToGray* (converted images to greyscale), *Smooth* (smoothed images to remove small artifacts), *Enhance Edges* (improved object identification), *IdentifyPrimaryObjects* (identified spheroids and excluded artifacts and border objects), *MeasureObjectSizeShape* (measured several size and shape features of each spheroid), *MeasureImageAreaOccupied* (measured total area in each image occupied by spheroids), *TrackObjects* (tracked each spheroid through sequential frames), and *ExportToDatabase* (generated CellProfiler Analyst properties file). Specific settings in each module were altered to optimise segmentation and analysis of spheroids of distinct characteristics. An example pipeline is provided on the Traject3d GitHub repository.

CellProfiler Analyst[18] (https://cellprofiler.org/, v2.2.0) was used to apply iterative, user-supervised machine learning to spheroids measurements in the resulting database. Specifically, PC3 spheroids were classified into bins based on their morphology: Round, Spread and Spindle. Machine learning to differentiate between these classes, using a maximum of 20 rules with Fast Gentle Boosting, and trained until accuracy was assessed using a confusion matrix (each class >90% accuracy in Supplementary Fig. 4c). Once generated, these classification rules were saved as a.txt file and directly imported into CellProfiler for classification without need for re-training after each acquisition of new data.

## Measuring and classifying 3D PC3 spheroids
We then imported the phase images acquired using the IncuCyte system into a second custom CellProfiler pipeline. This CellProfiler pipeline is similar to the one described above except that ExportToDatabase was replaced with three FilterObjects modules. We imported the classification rules generated in CellProfiler Analyst into each of these modules and filtered spheroids into Round, Spread and Spindle categories. Finally, we included ExportToSpreadsheet, OverlayOutlines and SaveImages modules in order to: a) export a data table containing classification, size, shape and movement measurements for each spheroid at each timepoint (saved as PhaseGrayCysts.csv in a sub-directory called Data), and b) to generate a binary image of spheroids outlines suitable for overlay onto the appropriate phase image downstream using KNIME (saved in PhaseGrayCystOutlines sub-directory). In the resulting data table, each table row is an object (spheroid) in the biological replicate, and each column is either

metadata (i.e. well, image filename, coordinates in image, classification) or a measurement of size, shape, and movement.

## Analysis of non-PC3 Cell lines

22 human prostate, breast, colorectal, canine kidney, and murine pancreatic cell lines (Supplementary Table 2) were cultured and imaged over multiple days. CellProfiler was used to segment the resulting images, track spheroids, and measure features of size, shape and movement. The resulting datasets of spheroids measurements were imported into a custom KNIME pipeline, in which they were merged with their corresponding experimental keys.

Correlation analysis to calculate Pearson correlation coefficient was performed, and results presented as a heatmap generated using the pheatmap[59] R package. Mean size, shape, and movement features were calculated per replicate (well), and PCA performed. Data were subsampled using GeoSketch[45], and t-SNE performed using the Rt-SNE[60] package. Subsampling depth and t-SNE parameters were as stated in respective figure legends. In the case of murine PDAC clones, PCA was performed on data averaged per experimental replicate. PCA and t-SNE results were plotted using ggplot2[61], and t-SNE point density was calculated using the MASS[62] package.

## Identification and quantification of phenotypes

We created a custom computational pipeline using KNIME Data Analytics Platform, enabling us to manipulate the large datasets generated by CellProfiler. This pipeline deals with analysis for both methods (user-defined and data-driven) of phenotype identification. The pipeline was built using the KNIME Analytics Platform[20] (https://www.knime.com/, v4.0.2) with R (https://cran.r-project.org/, v3.6.2) and Python (https://www.python.org/, v3.8) integrations. However, since analysis of the data presented in this work newer versions of R have been released; we have adapted and tested the analysis pipeline to use the newest version (v4.2.0) of the R software. Initially, the data is imported and then pre-processed, before reaching a branchpoint at which the analysis diverges to address each method. Thus, the user has a choice of running either of, or both, the branches depending on which method they wish to perform. Below we describe the different steps and outcomes of this analysis pipeline, in addition to the specifics of its use on the datasets presented in this work. Some parameters in the CellProfiler and KNIME pipelines can be adapted for use with other datasets. For further information and instructions, refer to the associated user manual provided in the Traject3d GitHub repository.

## Pre-processing

Combine experimental replicates and metadata: The data from each experimental replicate is imported and merged with its respective experimental key. This ensures that each well position is labelled with specific sample details. This merging is performed for each biological (experimental) replicate, before concatenating all replicates into one data table. We imported each experiment independently, before concatenating the resulting data tables.

Data filtering: The user can optionally filter the dataset by frame number, object lifetime (number of timepoints, e.g. hours, a spheroid was tracked for), or treatment. In our analyses, data acquired only up until 96 hours was utilised. This was based on the observation that some spheroids began to merge after this point. The resulting spheroids counts are shown in Supplementary Table 3.

Parse user-defined classifications: If user-defined classifications were generated in CellProfiler Analyst, and applied to a dataset, CellProfiler includes one binary column for each classification in the output data table. The pipeline parses these binary classification columns, and generates a single column containing the assigned classification (i.e. "round", "spindle", or "spread").

Tracking label correction: This method relies on temporal information about spheroid morphology, which enables us to identify recurring patterns of morphological change. For the analysis to work it is important that the tracking information is linear. However, when tracked objects split into two, or more, daughter objects, CellProfiler assigns the tracking label of the parent to all children. This means that downstream there can be multiple morphological tracks (spheroids) with the same label, complicating analysis. For this reason, we correct the tracking labels to ensure that each label corresponds to only one spheroid. To do this, in cases where a spheroid has split, we assign the parent label to the daughter spheroid with the largest area, and all other children (and their subsequent tracking events) are assigned a new label.

Normalisation: Finally, we calculate the Aspect Ratio (length/width) of the spheroids and normalise the dataset by Z-scoring each measurement column.

## Feature analysis

Spheroid feature quantification: We generate a heatmap of mean feature (area, shape, movement) over time, with statistical comparison between control and treatment groups. The user selects the features to be plotted, size of time interval for comparison, and which sample to use as reference. For this work, Z-score normalised data for spheroid area are presented in 12-hour time intervals as a heatmap generated using the ggplot2[61] R package. Mean spheroid area was calculated at each time interval per treatment. P values, comparing reference and non-reference samples, correspond to circle size (values as indicated in Figures). Statistical comparison was performed by Student's t-test, two-tailed, and a Bonferroni adjustment was applied to account for multiple testing. Experimental (n) and technical replicate numbers and the total number of spheroids imaged are stated in each Figure Legend, and in Supplementary Table 3.

PCA of replicates: For quality control purposes, two PCA analyses are performed, and results of each plotted using ggplot2[61]. In the first of the two, feature (area, Zernike moments, Displacement, DistanceTraveled) measurements are averaged such that each point in the PCA represents an individual well, and in the second this is done per experimental replicate.

Spheroid counts: The pipeline generates a line plot of normalised spheroid counts over time. Each treatment group is plotted as a separate line per experimental replicate and normalised to its respective initial count. An output of the initial counts is generated (CSV).

## User-defined phenotypes

Phenotype Quantification: We generate a heatmap of user-defined state frequency over time, with statistical comparison between control and treatment groups. The user selects the size of time interval for comparison, and which sample to use as reference. For this work, Z-score normalised data for user-defined state classification is presented in 12-hour time intervals as heatmaps generated using the ggplot2[61] R package. User-defined state classification is presented as a relative proportion of the total spheroids in a treatment, at a given time interval. P values, comparing reference and non-reference samples, correspond to circle size (values as indicated in Figures). Statistical comparison was performed on a contingency table of spheroid counts using the Cochran-Mantel-Haenszel test, which takes into account biological replicates. A comparison is only statistically significant where the effect was present across all biological replicates. We use the Breslow-Day statistic to test the assumption that the magnitude of effect of the treatment is homogeneous across all strata (biological replicates) – a non-significant p-value indicates homogeneity and is represented by a black dot in the heatmap. For this, we used the DescTools[63] R package. In both statistical tests, a Bonferroni adjustment was applied to account for multiple testing. Biological (n) and technical replicate numbers and the total number of spheroids imaged is stated in each Figure Legend, and in Supplementary Table 3.

Phenotype mean features: We calculate the means of all measured features for the user-defined states, which is output as a heatmap generated using pheatmap[42].

Representative outlines: To obtain a representative shape for each state, we first select the outlines of the 30 spheroids nearest to the group's mean morphological measurements. We then built on the approach outlined by Tweedy et al.[64] to select one of these objects as the group representative. Specifically, for each object: a predefined number of points were equidistributed along its two-dimensional boundary, the resulting boundary was expressed as a complex function of arc-length (with the real and imaginary parts corresponding to the $x$ and $y$ coordinates, respectively), and the power spectrum was computed. We then transformed all objects to a common scale by dividing the power spectrum of each one by its maximum value. To determine the object whose shape was most representative of a given cluster of objects, we performed principal component analysis on the associated set of scaled power spectra and selected the object that was closest (by Euclidean distance) to the global maximum of the estimated two-dimensional kernel density across the first two principal components.

Colour and overlay outlines: Binary (white on black background) images of spheroids outlines, as output by CellProfiler, were imported into KNIME, and segmented to identify individual spheroid outlines. We then compared the coordinates of the outline centroids, as calculated within KNIME, to those (of spheroids in the corresponding image) contained in the data table. This enabled us to link a spheroid state classification, be it user-defined or data-driven, with a specific outline. We then coloured the outlines by classification, recompiled the outline image, and overlaid it onto its corresponding phase image.

t-SNE: The analysis pipeline allows subsampling at a user-defined depth and method, random sampling versus Geometric Sketching (GeoSketch)[45]. t-SNE parameters are also user-defined, and elements of the resulting plot, such as point size and colour scheme, are adjustable. For the datasets in this work, subsampling and t-SNE parameters are as stated in respective figure legends. Generally, subsampling prior to t-SNE performed using GeoSketch, to ensure uniform sampling across phenotypic space. t-SNE was subsequently performed using the Rt-SNE[60] package, with PCA initialisation on the subset of non-redundant features listed in Supplementary Table 4.

## Data-driven phenotypes

Data-driven State identification: We first subsampled the dataset of PC3 and derivative lines using the Geometric Sketching[45] (GeoSketch) algorithm, to retain 20,000 objects. Using this algorithm enabled us to reduce computation time required for analysis, while retaining a subset of data evenly distributed across phenotypic space. We then applied the PhenoGraph[40] algorithm, as implemented in the cytofkit2[41] R package, on features listed in Supplementary Table 4, with a $k$-nearest neighbours value of 40. This subset was determined by selecting for non-redundant features based on cross-correlation analysis of preliminary datasets (Supplementary Fig. 1, Supplementary Table 4).

Trajectory Identification: Data are reconstructed into a temporal sequence of state classification, for each tracked spheroid. To ensure that we compare spheroids that existed from the start of imaging, we first filter tracked objects to exclude any without a tracked event at $t = 20$ hours, before trimming the sequences at the point at which tracking ceases for 50% of objects. We then remove objects with more than two consecutive missing values, and impute any remaining missing values by LOCF (Last Observation Carried Forward) in reverse using the imputeTS[65] R package. To allow us to cluster the spheroids based on their sequences of state classification, we convert the categorical (state classification) sequence data into 50 numerical dimensions by Multiple Correspondence Analysis (MCA) using the FactoMineR[66] package. Finally, we classify the tracked objects into

trajectories using PhenoGraph[40], as implemented in the cytofkit2[41] package, with $k$-nearest neighbours = 40, and generate a heatmap of results using the pheatmap[59] package. The resulting spheroid counts are shown in Supplementary Table 5.

Trajectory Motifs: A state frequency motif is plotted for each identified trajectory, using a user-specified time interval. We calculated state frequency at 12-hour time intervals and plotted frequency motifs using the ggseqlogo[67] R package, where state frequency is indicated by symbol (letter) height.

Trajectory Transition Plots: A two hour wide sliding window was used to calculate the frequency of transitions between states at each time interval, before compiling into 12-hour time intervals for plotting. Each interval was filtered to remove transitions that occurred at less than 2% frequency; this helps remove noise and generate more easily interpretable transition plots. The above was applied to each identified trajectory. Chord diagrams were plotted using the circlize[68] R package, whereas PCA transition plots were plotted using ggplot2[61].

Trajectory Representative Spheroid: For each trajectory, we first calculated state frequency and, in turn, the state with the highest frequency, at each timepoint. We refer to this as the consensus sequence. For each spheroid classified into a particular trajectory we then remove timepoints which are missing a classification and compare its sequence of state classifications to the consensus sequence, computing the (inverse) generalized Levenshtein distance using the base R adist function. This equates to a similarity score for each spheroid in comparison to the consensus sequence. The spheroid with the highest score is selected to represent the trajectory. Spheroid outlines were overlaid onto original phase images as described above.

Quantification of Data-driven State and Trajectory Classifications: We quantified state and trajectory classification, which we present as heatmaps generated using the ggplot2[61] R package. Values are presented as $\log_2$ fold change from reference sample. Dendrograms for the row and column order of Figs. 1h, 2e, 3b were computed using the complete linkage method for hierarchical clustering of the Euclidian distances. P values, comparing reference and non-reference samples, correspond to circle size (values as indicated in Figures). Statistical comparison was performed on a contingency table of counts using the Cochran-Mantel-Haenszel test, which takes into account experimental replicates. A comparison is only statistically significant where the effect was present across all experimental replicates. We use the Woolf statistic to test the assumption that the magnitude of effect of the treatment is homogeneous across all strata (experimental replicates) – a non-significant p-value indicates homogeneity and is represented by a black dot in the heatmap. We performed the Woolf test using the vcd[69] R package. Statistical tests for the quantification of data-driven states were Bonferroni adjusted. Experimental (n) and technical replicate numbers and the total number of spheroids imaged as stated in each Figure Legend, and in Supplementary Tables 3 and 5.

Colour and overlay outlines: As described for user-defined states.

## Reporting summary

Further information on research design is available in the Nature Research Reporting Summary linked to this article.

## Data availability

The RNAseq data from PC3 sublines used in this study are available in either the Short Read Archive database for PC3 E-cad + , GS689.Li in SRS354082), or the Gene Expression Omnibus for PC3-Epi, PC3-EMT14 in GSE48230. Unprocessed and uncropped western blot images are provided in Supplementary Fig. 19 and as Source data. Unprocessed imaging data supporting the findings of this study are available from the corresponding author on request. Due to the large size (-125GB) of the complete imaging data, large requests are anticipated to be reasonably answered within 4 weeks. Source data are provided with this paper.

## Code availability

All CellProfiler and KNIME analysis workflows, as well as a download link for sample data can be found in the Traject3d GitHub repository (https://github.com/davebryantlab/Traject3d)[70].

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

## Acknowledgements

This work was supported by the following grants; NIH K99CA163535 (D.M.B.), CRUK C596/A19481 (M.Na.), CRUK C7932/A25170 (K.N.), CRUK C596/A17196 and A31287 (E.S., AR.-F., L.M., A.H., M.Ne, D.S.), MR/L017997/1), (M.S) A17196, A25233, A29996 (J.P.M.), A29801 (C.M.), A22904 (H.Y.L.), IAA/CRUK (A26825 and A28223), O.J.S and T.RM.L. E.C.F. was supported by a University of Glasgow Industrial Partnership PhD studentship co-funded by Essen Bioscience, Sartorius Group. We thank M. Henry (University of Iowa) and K. Pienta for providing cell lines (Johns Hopkins Medicine International). We thank R. Carstens for ESRP1/2 shRNA plasmids. We also thank the Core Services and Advanced Technologies at the Cancer Research UK Beatson Institute with particular thanks to the Beatson Advanced Imaging Resource and Molecular Technologies. We thank the Life Science Editors Foundation for their support in preparation of this article as part of a Justice, Equity, Diversity and Inclusion (JEDI) Award.

## Author contributions

D.B. conceived the project. E.S., A.R.F, E.C., K.N., M.Na., and L.M. performed wet lab experiments under the supervision of D.B., E.F., A.H., D.S., and M.Ne. performed computational analyses under supervision of D.B. and C.M., J.M., M.S., O.J.S, T.RM.L. and H.L. provided materials. E.F. and E.S. composed the figures under supervision of D.B., E.F., E.S., and D.B. wrote the manuscript with input from all authors. E.C. and M.Ne, contributed equally. All authors discussed the results and commented on the manuscript.

## Competing interests

E.C.F. was supported by a University of Glasgow Industrial Partnership PhD scheme co-funded by Essen Bioscience, Sartorius Group. All other authors have no competing interests.
