## [Peer Review File · Nature Communications]

Reviewers' Comments:

Reviewer #1:

Remarks to the Author:

In Freckmann et. al., the authors image the real-time organoid morphology and motility for a set of prostate cancer cell lines and develop a method, named Traject3d, to profile the dynamics of the organoid morphology. The analytics model for organoids are in general lacking and more tools are needed. However, here are the concerns I have for the manuscript.

- As the methodology paper, the description of method does in general lack of rationale and details for choice of machine learning algorithm and parameters used. It's not clear to me how the 16 morphology states and 23 dynamic states are reached and what sample are used to reach this. The novelty of the workflow is also not clear. The authors using relative typical machine learning steps in processing the organoid morphology data and the machine learning based organoid morphology has been reported (PMID: 29593296). Perhaps, as the method paper, the dynamics states analysis is more novel, but it's not clear how robust/repeatability is the proposed organoid dynamics state clustering analysis in biological replicate and how consistent is the dynamics state transition pattern within each cluster.

- The paper is overall very descriptive, and lack of in-depth analysis or sufficient data to back the statement. For examples, authors states " (line 135)...PCA also allowed instances of batch effects to be identified (dashed lines; Supplementary Figure 2c; Supplementary Figure 3b), but failed to capture whether distinct phenotypes occur in parallel within a given cell line..."its not clear how authors compare PCA with tSNE and how PCA fails to capture distinct phenotypes based on the data analysis provided. "(line 140)... the distributions of which were conserved across experiments. This suggests that heterogeneity observed in 3D repeatedly occurs across experimental replicates, suggesting that this represents distinct co-occurring phenotypes rather than random variation." the distribution of the replicate in supp figure 2 e-f shows the replicate can be quite different from each others. More proper quantification (such as measuring the similarity or variability between replicate) should be performed to better support the claim.

- Some biological conclusions are also not fully supported by the reported data. For example, "line 385 ...depletion of ZEB1 or ESRP1/2 reduced or enhanced invasion in orthogonal invasion assays, respectively, confirming that these shape and motility changes represented bona fide changes to invasive potential..." the authors data shows correlation between morphology/motility to the invasion assays outcome modulating ZEB1 and ESRP1/2. To further confirm this morphology changes with invasion potential, some in vivo mouse model study should be included.

- Statistical concerns- Not clear what error bar represent in figure 1c, 2d; More specific details should be revealed on how the P-value is calculated for comparing the clustering results. Plots should be made to show the reproducibility of clustering results for the biological replicates. Author reported the total tracked organoid numbers in the graphs. This number seems to not reflect the number of independent organoids as the same organoids can be tracked over times and hence would be counting multiple times. Perhaps the total frame number should be also provided in the graph.

- "line 192 ... increase in Round state from other metastatic samples or a single round of trans endothelial migration (TEM2-5, GS672.Ug, JD1203.Lu). The PC3-EMT sample was the only instance of induction of dual behaviours (Spread and Spindle)..." In the figure 1h, It appear JD1203.LU is further distance than rest of cells based on the dendrogram on figure 1h (even more so tha PC-3 EMT which are closer to PC3- EPI and Ecad+). This seems to be inconsistent with the statement and need more clarification.

- Though the dynamics state analysis could be novel, its not clear this analysis derive new insight outside then static morphology analysis. The dendrogram plot from static morphology clustering and dynamics morphology clustering (figure 2E and figure 3b) shown the PC -3 derived cells are generally classify the same way. Not sure how much added value the dynamics morphology analysis can provides.

- "Line 337... invasive cells from both cell pairings (PC3-EMT, GS689.Li), with the master transcriptional regulator of EMT ZEB1 strongly induced in both of these cells38 (Figure 4c-d). Accordingly, shRNA-mediated depletion of the transcriptional repressor ZEB1..." not clear how all

others genes expression are different in the author selected pairs and not sure how the ZEB1 was identified among them. More data and rationale should be provided.

- "line 339 Accordingly, shRNA-mediated depletion of the transcriptional repressor ZEB1 in PC3-EMT cells resulted in restoration of expression of 20% of the top epithelioid related genes identified from the epithelioid cells in each pairing (Figure 4a,b,e)...." Not clear how this statement is reached, it also lack of experimental description of knockdown experiment. How is the gene expression been accessed in the knockdown sample?

Reviewer #2:

Remarks to the Author:

I think the authors had an extremely cool idea here, and I fully support this sort of unbiased phenotype discovery via high content imaging. I dearly wish that I found the authors' conclusions credible, but unfortunately I do not.

In looking at some of the images and movies presented in the figures and the supplemental movies, I was troubled by a few aspects.

Firstly, cell density- looking at e.g. Supplemental Figure 3A - some of these lines are dense to the point that in some cases more than half of the field of view or essentially the entire field of view is identified as a single object, per the yellow outlines (and confirmed by the authors' CellProfiler pipeline's settings- any cells that touch are considered a single structure). Yet somehow the authors say they are getting quantitative measurement data on things like distance travelled and movement from these data sets (the figure captions for Supplemental 3 b-c confirm it is for the same data sets in 3 a) - there is simply no possible way that any meaningful information could be extracted from images that look like that with those analysis settings.

In watching some of their example movies as well, it seemed to me that the "cell states" changes that they were trying to demonstrate were caused by two neighboring cells coming together and then splitting back apart. See Figure 5 k and Supplementary movies 14 and 19. The authors describe that in the caption of Figure 5 as "adoption of non-round shape", but it's clearly not a cell adopting a shape but multiple cells aggregating. If that was happening in the selected data, what was going on otherwise?

This led me to run the authors' pipeline on the provided data, using the same CellProfiler version - I confirmed via Python that my quantitative results were identical to theirs re: tracking. The one change I did make was to enable the creation of the "tracking image" to show exactly what each tracked object was, and to add an extra SaveImages module to save that image out. I've attached the results from one timelapse (Experiment 2 Plate 1 D7_1) here as a movie - as you can see, what is considered "object 16", "object 31", "object 56", "object 68", and "object 88", are in fact several independent objects moving on independent trajectories, sometimes together and sometimes apart, and that by definition must be affecting the measurements of sizes and of trajectories/displacements frame-to-frame. The authors say that when multiple sub-objects get the same label, they automatically give it only to the largest one, but in the case of things splitting in this way (going back and forth between one and many objects and sometimes splitting in very different ways) there is no possibility that it properly corrects for the error. I tried to check the results of this "correction" myself in the KNIME software but I could not get even the first node to execute on my machine (it failed somewhere between "Only keep PhaseGrayCysts" and "Make DataFile Filenames variables") so I could not verify. It seems in my survey of the tracked output that at least a quarter of the fields of view, if not more, suffer from this issue, and the ones that don't have mostly-round cells that mostly sit more or less in place for the whole time of imaging.

I unfortunately therefore have a very difficult time believing that with such unclean data going in that any real conclusions can be drawn with respect to changes in shape over time or trajectory over time for individual cells. Tracking is a uniquely difficult problem in image analysis because it requires EXTREMELY high-fidelity segmentation (something that CellProfiler is notoriously poor at with brightfield data, though the analyst has done an excellent job under those conditions) and/or incredibly laborious hand-cleaning of the data. I believe that in general the things that they think

are motile probably usually are, and that things they think aren't motile probably usually aren't, and to a certain density point on the plate "round" vs "spindle" vs "spread" is probably fine (although the fact that the analyst left a metric of cell position, Location_Center_Y, as one of their rules doesn't indicate a high familiarity with what features one would want to leave into a final rules list in a putatively location-agnostic phenotype of interest), but beyond that I simply can't trust any single-cell measurements of sub behaviors of "trajectory" and possibly also not of "state" (since state includes a metric of motility).

I think if the caveats are made much, MUCH clearer and the conclusions are scaled way, WAY back, this is a really great idea. Even just saying that you're looking at an IMAGE level at a) distribution of shape changes over time and b) distribution of trajectories/displacements (basically, % fast vs slow cells) found over time, WITH appropriate caveats re: density and segmentability, I think you could find some cool phenotypes here and do some cool biology. But as currently written it's far too over-interpreted for the quality of the data and I cannot support its publication at this time.

(You really also should have in the supplemental material your whole western blots, with loading controls for EVERY blot not just representative ones!)

Reviewer #3:

Remarks to the Author:

Freckmann and Sandilands et al describe the computational method Trajectory identification in 3D culture (Traject3d) to analyse live 3D cell models. The method uses the widely available Incucyte imaging system so could be widely adopted by researchers.

This reviewer is not qualified to critique the computational elements of the paper and leaves this area to other reviewers.

The authors train the method on several cell lines that appear to have been grown as planer (2.5D) clumps in Matrigel – which in my experience is quite an unusual cancer model (but maybe the authors can explain why they are relevant). Typically, if researchers decide to use more complex 3D cultures over 2D cultures, it is to model a phenotype that can only be achieved in in 3D (e.g. self-organisation, differentiation, nutrient gradients, invasive migration etc). While the computational imaging methods may be impressive, the lack of a more biomimetic cell culture model will limit the impact of the work.

The manuscript is well written but comprises lots of small observations that I struggled to collectively summarise once I had finished reading. The figures are nicely presented if slightly difficult to understand on first viewing.

Major Points

1) The 3D cultures used in the manuscript are extremely simple and lack much of the complexity of contemporary biomimetic models – notably spheroids and organoids. The cell lines may be appropriate for building the initial tools, but greatly limits the real-world use of Traject3d. Can the authors demonstrate the method working on a more commonly used 3D model such as spheroids or organoids? If the method can only work on clumped cell lines this will limit the application and should be clearly stated in the abstract. Currently the abstract specially mentions organoids but I could not find any evidence of them in the main text of the manuscript.

2) One of the primary uses of single-cell analysis is to resolve different cell types. Can Traject3d be used to perform cell-type-specific analysis of co-cultures? For example, in the PC3-EMT model, is data being produced for both PC3 cells and macrophages? Demonstration of co-culture analysis would greatly increase the usefulness of the method.

Point-by-Point Response to Reviewers

Reviewer comments are presented in bold. Our responses, non-bold. We number the reviewer points to allow for easier referencing of shared comments. In the manuscript, changes to text are indicated in blue font.

Reviewer#1

In Freckmann et. al., the authors image the real-time organoid morphology and motility for a set of prostate cancer cell lines and develop a method, named Traject3d, to profile the dynamics of the organoid morphology. The analytics model for organoids are in general lacking and more tools are needs. However, here are the concerns I have for the manuscript.

- 1. As the methodology paper, the description of method does in general lack of rationale and details for choice of machine learning algorithm and parameters used.**

In a quest for brevity, we previously overedited the manuscript. We have added substantial rationale and explanation at key points through the manuscript. Although now a much-lengthened manuscript, we thank the reviewer for prompting us to add in these important points as we believe the manuscript and our rationale is now much easier to understand.

The text was updated for clearer rationale description. This includes:

- Software choice, p. 4
- Machine learning (user-defined), p. 8
- Machine learning (data-driven), p. 11
- Parameters used for data analysis, p. 11

We describe the main points here briefly:

- *For image analysis:* Our choice is based on accessibility and ease of use. CellProfiler, CellProfiler Analyst and KNIME are open-source freeware that can be used by biologists without requirement for coding skills.
- *Machine learning algorithm (to generate user-defined classifications):* We wanted:
 1. an in-built module in CellProfiler Analyst for simple click-and-classify approaches that CellProfiler Analyst provides, that
 2. worked effectively (as evidenced by 91-97% fidelity of model prediction to user-defined training; Supplementary Figure 4c), and
 3. a classification model that, after training, can be converted into 'rules' that are exported and applied to new data. The key here is to allow a user to apply these training rules to new data as it is acquired. Without this, the user needs to wait until all data is collected before applying the training.

A 'Fast Gentle Boosting' model met these criteria.

- *Machine learning algorithm (for data-driven analysis):* Subpopulations in the data are identified using the PhenoGraph algorithm which is widely used across multiple fields for similar purposes. We selected this over other clustering algorithms (ClusterX, DensVM, FlowSOM, k-means) as it allowed unsupervised clustering into subpopulations without reliance on prior dimensionality reduction (such as t-SNE).
- *Parameters utilised for data analysis:* We wanted a label-free, generalisable approach for analysing 3D behaviour to enable wide usability. We use non-redundant features generated by CellProfiler to minimise computation time (all listed in Supplementary Figure 1; non-redundant subset listed in Supplementary Table 4). Size is defined by Area, shape via Zernike polynomials (a widely used system where collectively the

'Zernike moments' represent shape properties with no overlapping or redundant shape information), and movement features generated by CellProfiler. We exclude texture features as these are affected by variations in the focal plane of imaging.

2. Its not clear to me how the 16 morphology states and 23 dynamic states are reached and what sample are used to reach this.

We have clarified how the states were identified (p. 11). On p. 14, we have clarified how the trajectories (dynamic states) were identified. We have also updated the relevant subsections of the Methods to include more elaboration and explanation of rationale.

The Methods section under the heading 'Data-driven phenotypes' (starting p. 41) includes individual method subsections for:

- Data-driven State identification
- Trajectory Identification
- Trajectory Motifs
- Trajectory Transition Plots
- Trajectory Representative Acinus
- Quantification of Data-driven State and Trajectory Classifications

The samples that underpin the states and trajectories are listed as part of Supplementary Figure 10, under the 'Sample' heading. This includes samples and their controls for PC3 a) metastasis variants, b) ZEB1 alteration, c) ESRP1/2 alteration, d) HGF alteration and e) PC3M docetaxel-resistance series. Analysis of these samples together allows direct comparison of states/trajectories.

An important point is that the above samples were similar enough in their morphogenesis characteristics that image sequences of the same length (e.g. 96h) were compatible. This is essential as morphogenesis that is vastly different between samples is incompatible with analysis over the same length of time (for example slow-growing spheroids vs objects growing/invading quickly that 'overtake' the image). We explicitly state this point on p. 11.

For the remaining cell lines (from Supplementary Figures 2-3), such differential behaviours across time scales were incompatible with using the same imaging timing (i.e. some samples merge quickly precluding analysis from that point). Accordingly, these were not included in the PC3 analysis. These would need their own comparison across a time frame appropriate for their rate of morphogenesis.

3. The novelty of the workflow is also not clear. The authors using relative typical machine learning steps in processing the organoid morphology data and the machine learning based organoid morphology has been reported (PMID: 29593296).

We have clarified these points in the text (p. 4) including explicitly stating and referencing where our approach is similar to other reported approaches.

Our advance is to analyse 3D structures over time (as the reviewer identifies below). We are not the first to analyse 3D structures from label-free imaging, but our work is different from other analyses in the two following important ways:

In 3D culture, multiple phenotypes can exist concurrently within a sample. Typical analyses average features across the entire sample. This results in an aggregate which may not accurately represent distinct, co-occurring phenotypes. We provide a way to disaggregate this

by identifying distinct subtypes. We acknowledge that OrganoSeg (the PMID the reviewer lists above) can perform a similar function on static images yet lacks the functionality to present what each subtype means that our approach provides.

The essential feature that OrganoSeg and other similar approaches do not address – *which Traject3d does* - is that morphology changes over time. Sampling a singular timepoint fails to capture the dynamics of morphogenesis. For instance, we find that alternate phenotypes can be comprised of the same morphology states but use them in differing ways. Two objects that exhibit different phenotypes (appreciated only because they were imaged over time) may at certain timepoints appear indistinguishable. Without live imaging and the analysis we provide, static timepoint analysis underestimates the phenotypic heterogeneity present.

4. Perhaps, as the method paper, the dynamics states analysis is more novel, but it's not clear how robust/repeatability is the proposed organoid dynamics state clustering analysis in biological replicate and how consistent is the dynamics state transition pattern within each cluster.

We have updated the Results section of the manuscript to make clear that the methods used for statistical comparison take consistency across biological replicates into account. This is initially done in detail at the top of p. 10 for the user-defined states and re-emphasised later when discussing subsequent results (such as quantification of data-driven states and trajectories) (see p. 13 and p. 16).

A crucial point is that the states/trajectories that we identify are robustly and repeatedly identified across independent experimental replicates. Our analysis already provided such quantitation throughout. Variation within each subtype over time is presented in Supplementary Figure 10.

Figure 2d shows, as an example of a single cell type (PC3), the mean frequency of classification of each subtype, across time, along with the standard error of the mean across independent experiments. This shows each state occurs with remarkable repeatability in a given sample across biological replicates (independent experiments).

Throughout the manuscript we use a statistical method that takes into account consistency across biological replicates, the Cochran-Mantel-Haenszel test, for both user-defined and data-driven states and trajectories. This is displayed throughout the manuscript as heatmaps which quantify changes relative to a control sample (e.g. Figure 1h for user-defined; Figures 2e and 3b for data-driven). On top of this, we provide further statistical analysis (Breslow-Day and Woolf tests) to indicate consistent magnitude of effect.

The consistency of transition dynamics between states is depicted through line colour used in the plots (Figures 3c-e, 5j,k, 6f,g; Supplementary Figures 12A-C), which represents the frequency at which the state transition occurred.

Therefore, our methodology already consists of robust statistical quantitation. We also provide a reviewer-only figure (at end of this file) to show the individual replicates for both cell states and trajectories. This shows the remarkable consistency across replicates and conditions. We feel that depicting the significance (using the above tests) in the main figures, rather than complicating it with displaying every replicate in the main figure, is a superior method of presentation.

5. The paper is overall very descriptive, and lack of in-depth analysis or sufficient data to back the statement.

We strongly disagree that our analysis is descriptive and lacks depth. In addition to identifying heterogeneous phenotypes, we identify an extensive molecular pathway controlling the formation of invasive phenotypes. This includes identifying a ligand (HGF), its receptor (c-Met), a downstream effector of the transcriptional regulator ZEB1, as well as a key target of ZEB1 in the splicing factors ESRP1/2. Moreover, we identify that co-targeting this pathway can restore sensitivity to otherwise docetaxel-resistant tumour cells.

We have updated the introduction of the article to include a summary of these key points (p. 4).

6. For examples, authors states ” (line 135)...PCA also allowed instances of batch effects to be identified (dashed lines; Supplementary Figure 2c; Supplementary Figure 3b), but failed to capture whether distinct phenotypes occur in parallel within a given cell line...”its not clear how authors compare PCA with tSNE and how PCA fails to capture distinct phenotypes based on the data analysis provided.

We see how this statement was confusing. We have extensively rewritten this section, now p. 7.

We clarify that our intended meaning was that PCA and tSNE were alternate approaches to display global data (the size, shape, movement features from each cell line under observation). This allowed us to see cell lines that shared similar characteristics, such as grouping cell lines together that displayed spindly 3D phenotypes. It also allowed us to see instances of batch effects. However, PCA was not helpful in distinguishing cell lines where heterogeneous distinct phenotypes were co-occurring in a sample.

7. “(line 140)... the distributions of which were conserved across experiments. This suggests that heterogeneity observed in 3D repeatedly occurs across experimental replicates, suggesting that this represents distinct co-occurring phenotypes rather than random variation.” the distribution of the replicate in supp figure 2 e-f shows the replicate can be quite different from each others. More proper quantification (such as measuring the similarity or variability between replicate) should be performed to better support the claim.

We have updated the text to highlight where batch effects occur and to make clear that the figures that the reviewer refers to above were for the purposes of quality control (p. 7).

We emphasise that we provided robust quantitation of the frequency of subtype appearance across biological replicates. Please see response to your point 4 for further explanation.

Supplementary Figure 2e-f is specifically included to make the point, as the reviewer states, that replicates can be quite different from one another (for example, RWPE-1 experiments in Supplementary Figure 2e have a clear batch effect, particularly in biological experiment replicate 2). We use this as a simple visual quality control demonstration before the robust statistical analysis described above.

Please note that because of such batch variation across experiments, RWPE-1 cells were not analysed further throughout the work. PC-3 are extensively quantified in the main figures.

8. Some biological conclusions are also not fully supported by the reported data. For example, “line 385 ...depletion of ZEB1 or ESRP1/2 reduced or enhanced invasion in

2022 Nat Comms Freckmann Response to Reviewers
orthogonal invasion assays, respectively, confirming that these shape and motility changes represented bona fide changes to invasive potential...” the authors data shows correlation between morphology/motility to the invasion assays outcome modulating ZEB1 and ESRP1/2. To further confirm this morphology changes with invasion potential, some in vivo mouse model study should be included.

Our assays involve 3D structures in matrix gels (Matrigel). Therefore, where we observe movement within these 3D movies, this is bona fide invasion. We cross-validate this in secondary assays (invasion of cells into matrix from a wounded monolayer). These represent two, independent, frequently used assays for invasion.

That ESRP1/2 and ZEB1 can be involved in invasion is well-documented in the literature. Our advance here is identifying that a subpopulation of cells have elevated HGF expression such that it drives an ZEB1-ESRP module to regulate motility.

We assert that asking for *in vivo* mouse work is arduous and outside the scope of a methodology for analysing 3D culture behaviours. However, we are happy to amend the language used and we now refer to motility instead of invasion (see p. 19).

9. Statistical concerns- Not clear what error bar represent in figure 1c, 2d;

We have updated Figure 1c and 2d legends to include this missing information. We thank the reviewer for pointing out that this information was missing. The error bars represent the standard error of the mean across biological replicates. This is clarified in the manuscript on p. 52. Please also see our response to your point 4.

10. More specific details should be revealed on how the P-value is calculated for comparing the clustering results.

A description on how statistics were performed was provided in “Quantification of Data-driven State and Trajectory Classifications” (p. 43). We have updated the Methods subsection reference above to provide further clarification – namely that the statistical tests are performed on a contingency table of spheroid counts – and explanation that these tests take biological replicate into account.

11. Plots should be made to show the reproducibility of clustering results for the biological replicates.

We assert that we provided a more appropriate demonstration of reproducibility; the data is always presented in heatmaps where circle size is representative of p-value for comparison across biological replicates (Cochran-Mantel-Haenszel test), and further demonstration of consistent magnitude of effect (black dot, Breslow-Day and Woolf tests).

We feel that our existing approach is a superior method for the central point: demonstrating that what we find represents trends repeatedly occurring (or not) across distinct biological experiments. See also the reviewer-only figure accompanying your point 4 above.

12. Author reported the total tracked organoid numbers in the graphs. This number seems to not reflect the number of independent organoids as the same organoids

2022 Nat Comms Freckmann Response to Reviewers
can be tracked over times and hence would be counting multiple times. Perhaps the total frame number should be also provided in the graph.

We thank the reviewer for their keen observation, which highlights that this needed clarification. To make this clearer, we have updated the figure captions to reference Supplementary Table 5 (which lists number of segmented spheroids after filtering steps, as well as the number of tracked spheroids). For consistency, the figure captions for all other figures have also been updated to reference their respective supplementary tables of spheroid counts.

13. “line 192 ... increase in Round state from other metastatic samples or a single round of trans endothelial migration (TEM2-5, GS672.Ug, JD1203.Lu). The PC3-EMT sample was the only instance of induction of dual behaviours (Spread and Spindle)...” In the figure 1h, It appear JD1203.LU is further distance than rest of cells based on the dendrogram on figure 1h (even more so tha PC-3 EMT which are closer to PC3- EPI and Ecad+). This seems to be inconsistent with the statement and need more clarification.

It was not our intent to imply that the sample grouping stated in the text was based on the dendrogram in Figure 1h. We have clarified in the text that the above was an observation and now clearly highlight that JD1203.Lu is contrastingly distanced in the dendrogram (*p. 10*).

14. Though the dynamics state analysis could be novel, its not clear this analysis derive new insight outside then static morphology analysis. The dendrogram plot from static morphology clustering and dynamics morphology clustering (figure 2E and figure 3b) shown the PC -3 derived cells are generally classify the same way. Not sure how much added value the dynamics morphology analysis can provides.

We respectfully disagree with the reviewer on this point for two reasons:

A. The dendrogram clustering in Figures 2e and 3b is not the same. As the reviewer highlighted earlier, JD1203.Lu is an outlier in the former, whereas in the latter it clusters with GS672.Ug and TEM2-5.

B. It is *expected* that quantification of cell state frequency would bear similarity to that of the trajectories. The trajectories are built from the cell states, ordered in sequence from live imaging. The ‘static’ analysis required live imaging to collect these multiple timepoints to ensure that we capture the full repertoire of states. One could, in theory, only focus on the static shape analysis, but a deconstructed view of the data cannot alone capture how states may transition from one to another over time to form distinct phenotypes.

The new organoid imaging figure we provide (Supplementary Figure 18) shows that while certain treatments (TGFB1, IL6) would seem to induce changes in rarer states, when only examining the states stripped of time, reconstituting these back in temporal order (the trajectories) showed that such changes were inconsequential for the phenotypes detected.

When the appearance of a heterogeneous state always and only gives rise to a distinct phenotype but not another, it may be sufficient to simply look at frequency of one state versus another and predict phenotypes. However, our data would tell us that other considerations are needed. Much like the same set of six letters can be used to spell incongruous words such as ‘listen’ and ‘silent’, identifying the frequency of distinct letters (or cell states) alone would not allow one to understand how these are used in order to give rise to distinct words (or trajectories/phenotypes). Similarly, the organoid experiments indicate that in some cases

heterogeneity in cell states exists, but that these may be inconsequential to the resulting phenotype. For example, 'colour and color' have heterogeneity that could be detected simply by identifying the frequency of letters (cell states), but that this does not give rise to altered meaning/phenotype. Moreover, heterogeneity not detected by static frequency alone can still exist, such as 'meter and metre' or 'centre and center', but similarly not affect the final (meaning) phenotype once temporally ordered.

This is what sets our approach apart from other excellent approaches, including OrganoSeg that the reviewer mentions above: a static picture of the states is not enough, it's how they are used in order that matters. Traject3d is the first approach to do this for 3D culture that we are aware of.

We have clarified this on p. 25.

15. "Line 337... invasive cells from both cell pairings (PC3-EMT, GS689.Li), with the master transcriptional regulator of EMT ZEB1 strongly induced in both of these cells38 (Figure 4c-d). Accordingly, shRNA-mediated depletion of the transcriptional repressor ZEB1..." not clear how all others genes expression are different in the author selected pairs and not sure how the ZEB1 was identified among them. More data and rationale should be provided.

We have amended the manuscript (p. 18) to clarify this point. Briefly, as EMT was noted in the biological processes identified in Figure 4c, we queried the changes in EMT-related transcriptional regulators. This revealed that ZEB1 was the most altered EMT-related transcription factor between cell pairs 1 and 2 (this was presented as Figure 4d).

16. "line 339 Accordingly, shRNA-mediated depletion of the transcriptional repressor ZEB1 in PC3-EMT cells resulted in restoration of expression of 20% of the top epithelioid related genes identified from the epithelioid cells in each paring (Figure 4a,b,e)...." Not clear how this statement is reached, it also lack of experimental description of knockdown experiment. How is the gene expression been accessed in the knockdown sample?

We have amended the text, on p. 18, to clarify this point.

Briefly, this data is mined from RNAseq analysis. First, the Top 50 epithelioid or EMT-associated genes were calculated as those with highest differential expression in common between epithelioid (PC3-Epi, PC3-E-cad+) versus mesenchymal (PC3-EMT, GS698.Li) cell pairings. This is presented as the heatmap on the left in Figure 4e.

Second, ZEB1 is typically a repressive transcription factor repressing transcripts associated with epithelial identity. We therefore compared the previously identified Top 50 epithelioid genes to RNAseq data from control *versus* ZEB1-depleted (two independent clones) mesenchymal PC3-EMT cells. This revealed that 20% of 'Top 50 epithelioid genes' had their expression substantially restored (often 10-fold on a Log2 scale) after ZEB1 shRNA, consistently across both knockdown clones. This is presented as asterisks on the heatmap on the left of Figure 4e, and those in red text on the expanded heatmap on the right of Figure 4e.

Reviewer #2 (Remarks to the Author):

- 1. I think the authors had an extremely cool idea here, and I fully support this sort of unbiased phenotype discovery via high content imaging. I dearly wish that I found the authors' conclusions credible, but unfortunately I do not.**

We thank the reviewer for their general support of our concept. We hope that our clarifications below will show that points raised have been addressed.

- 2. In looking at some of the images and movies presented in the figures and the supplemental movies, I was troubled by a few aspects.**

Firstly, cell density- looking at e.g. Supplemental Figure 3A - some of these lines are dense to the point that in some cases more than half of the field of view or essentially the entire field of view is identified as a single object, per the yellow outlines (and confirmed by the authors' CellProfiler pipeline's settings- any cells that touch are considered a single structure). Yet somehow the authors say they are getting quantitative measurement data on things like distance travelled and movement from these data sets (the figure captions for Supplemental 3 b-c confirm it is for the same data sets in 3 a) - there is simply no possible way that any meaningful information could be extracted from images that look like that with those analysis settings.

We have clarified these points, and included rationale, in the text (p. 9).

At the outset, we clarify that we do not include any images for analysis that consist of large, merged objects (or have other problems due to segmentation). Rather, we limit our analysis to a time interval where objects can be appropriately detected across all samples being compared. Practically, this means only ever comparing images from different samples up to the same timepoint BEFORE such merging begins. If this is not possible, then these samples are not compared.

We point out that the fact that objects merge at later timepoints, and are subsequently considered one object, doesn't preclude earlier measurements from being meaningful. One would simply need to stop analysis at a timepoint before which this event occurs, and the preceding data are valid.

Supplementary Figure 3a is provided as an extreme example of how, from even a single image timepoint (72h), clones of the same genotype exhibit vastly different phenotypes (i.e. emphasising the heterogeneity in 3D that is the main target of our approach). These samples were not further analysed using Traject3d.

For all subsequent work in the manuscript we filter to only include data up to the point before which objects start merging (stated in "Data filtering" on p. 38) in order to avoid the analysis and quantification of large aggregates.

We extensively and explicitly point this out in the manuscript on p. 9 and p. 14.

- 3. In watching some of their example movies as well, it seemed to me that the "cell states" changes that they were trying to demonstrate were caused by two neighboring cells coming together and then splitting back apart. See Figure 5 k and Supplementary movies 14 and 19. The authors describe that in the caption of Figure**

5 as "adoption of non-round shape", but it's clearly not a cell adopting a shape but multiple cells aggregating. If that was happening in the selected data, what was going on otherwise?

We clarify that we did not select the images and movies provided as the best of a bad bunch. Our approach selects these images and movies computationally in an unbiased manner as the most representative, as described in the manuscript (see "Trajectory Representative Spheroid" on p. 42). We have updated the text, on p. 15, to make clear that these representative spheroids were selected computationally in an unbiased manner.

Some phenotypes, such as non-round states and merging events, can result in a similar shape/outline (see state K in Supplementary Figure 6). See also new data provided imaging organoids (Supplementary Figure 18) that shows detection of air bubbles or instances where organoids touch. We strongly argue that dealing with such events should be part of the data and analysis as these are bona fide events that can and do happen in 3D culture. Users of our approach will also need to deal with these events. Our intention is for Traject3d to provide an honest capture of what happens and let the user interpret biological significance (e.g. air bubbles). We have updated the manuscript to clarify this point on p. 12.

Supplementary Figure 6 shows 10 representative spheroids (of 30) that are automatically output per cell state. This shows that the majority of objects are not merging spheroids. New Supplementary Figure 18 shows that air bubbles and merging objects are extremely rare events in organoid culture. The user, therefore, has the option of interpreting any given cell state in context of the most representative spheroids, allowing them to decide whether a state is a majority merging objects or whether this is a rare event that nonetheless has a co-incident shape to a bona fide non-merging cell state. In either case, this is information that allows the user to interpret the data, rather than something that should be selectively excluded. That these merging events or air bubbles 'make it through' is exactly the point of transparent collection of all of the events that occur, rather than something that should cause alarm as they should have been filtered these out (in a biased fashion). Our approach is honest, open, transparent data collection and presentation.

We now emphasise this important point in the manuscript extensively and explicitly (see p. 14 and p. 28).

4. **This led me to run the authors' pipeline on the provided data, using the same CellProfiler version - I confirmed via Python that my quantitative results were identical to theirs re: tracking. The one change I did make was to enable the creation of the "tracking image" to show exactly what each tracked object was, and to add an extra SavImages module to save that image out. I've attached the results from one timelapse (Experiment 2 Plate 1 D7_1) here as a movie - as you can see, what is considered "object 16", "object 31", "object 56", "object 68", and "object 88", are in fact several independent objects moving on independent trajectories, sometimes together and sometimes apart, and that by definition must be affecting the measurements of sizes and of trajectories/displacements frame-to-frame. The authors say that when multiple sub-objects get the same label, they automatically give it only to the largest one, but in the case of things splitting in this way (going back and forth between one and many objects and sometimes splitting in very different ways) there is no possibility that it properly corrects for the error. I tried to check the results of this "correction" myself in the KNIME software but I could not get even the first node to execute on my machine (it failed somewhere between "Only keep PhaseGrayCysts" and "Make DataFile Filenames variables") so I could not verify. It seems in my survey of the tracked output that at least a quarter of the fields of view,**

if not more, suffer from this issue, and the ones that don't have mostly-round cells that mostly sit more or less in place for the whole time of imaging.

I unfortunately therefore have a very difficult time believing that with such unclear data going in that any real conclusions can be drawn with respect to changes in shape over time or trajectory over time for individual cells. Tracking is a uniquely difficult problem in image analysis because it requires EXTREMELY high-fidelity segmentation (something that CellProfiler is notoriously poor at with brightfield data, though the analyst has done an excellent job under those conditions) and/or incredibly laborious hand-cleaning of the data. I believe that in general the things that they think are motile probably usually are, and that things they think aren't motile probably usually aren't, and to a certain density point on the plate "round" vs "spindle" vs "spread" is probably fine (although the fact that the analyst left a metric of cell position, Location_Center_Y, as one of their rules doesn't indicate a high familiarity with what features one would want to leave into a final rules list in a putatively location-agnostic phenotype of interest), but beyond that I simply can't trust any single-cell measurements of sub behaviors of "trajectory" and possibly also not of "state" (since state includes a metric of motility).

I think if the caveats are made much, MUCH clearer and the conclusions are scaled way, WAY back, this is a really great idea. Even just saying that you're looking at an IMAGE level at a) distribution of shape changes over time and b) distribution of trajectories/displacements (basically, % fast vs slow cells) found over time, WITH appropriate caveats re: density and segmentability, I think you could find some cool phenotypes here and do some cool biology. But as currently written it's far too over-interpreted for the quality of the data and I cannot support its publication at this time.

We break the reviewer's points down into subsections.

Data loading

We sincerely apologise that the reviewer was unable to run the analysis pipeline. We tested the provided pipeline, sample data, and instructions across multiple computers and users and did not encounter the same problem. We attempted to replicate the error. We can only do this if the required data files are missing from the directory imported into the pipeline. We would be more than happy to provide support to run the analysis (however this may work across peer review).

Segmentation

We completely agree that accurate segmentation is a central requisite (and challenge!) in image analysis. We thank the reviewer for assessment that we have provided an 'excellent job' of segmentation using CellProfiler. That this can be done extremely well in CellProfiler for 3D imaging is the reason that Traject3d is based on CellProfiler's output.

We have adjusted the text, p. 28, to suggest alternate methods for image segmentation and tracking, that may perform better for more challenging image sets, and how these can be used alongside CellProfiler to generate the data required for Traject3d.

'Location_Center_Y' in user-defined rule set

This was (surprisingly) automatically generated as one of twenty features in our training set. As we did not see any effect of this on our training (no positional bias in classification of objects), we chose not to manually curate the definition rules. We can remove this feature and re-run the data, if appropriate.

Tracking

The reviewer points out that multiple independent objects can have the same tracking ID if they touch at some point. What the reviewer refers to is the 'parent object' tracking ID, but each of these also has a 'child object' tracking ID. Whilst this is useful in CellProfiler, for instance, when tracking dividing cells into daughter cells it is problematic for tracking spheroids that may touch. Our tracking label correction in the method addresses a lack of tracking label assignment options in CellProfiler, enabling us to ensure that tracking labels are unique to a single tracked object.

In the case of distinct cell states, this analysis is agnostic to the tracking labels as it makes use of unique object IDs. While most features used to define cell states (size, shape) will be completely unaffected, there may be some contribution of imperfect tracking to motility features. Therefore, we may underestimate some motility contribution to cell states.

In the case of the trajectories, we need to compare objects that have been continuously tracked for the exact same period of time. We filter out any tracked objects that do not meet this criteria. When objects cyclically merge and split, the shape often changes substantially, causing CellProfiler to assign the 'new' objects a fresh tracking label. This can be seen in the reviewer's example ('object 16', 'object 31', 'object 56', 'object 68', and 'object 88'). It is essential to emphasise here that these 'new' objects, as defined by having a new label, do not exist long enough to make it through these filtering steps we perform when identifying trajectories (see 'Trajectory Identification' on p. 41). Therefore, these are not affecting trajectory analysis.

We must also point out that correcting the tracking label to account for objects repeatedly merging and splitting, and ensuring a consistent tracking label, would not address that outlines in certain frames are comprised of multiple acini touching. These would generate a singular object that is an aggregate of what are supposed to be two independent tracked objects. These merged outlines would remain in the analysis, and consequently as part of the identified trajectories, regardless of the correction performed. This is incompatible with requiring comparison of distinct tracked objects to identify distinct trajectories. Therefore, correcting the tracking as such would not fix the central biological issue of merging and splitting.

All of these are excellent points that we had addressed with our approach, but – and we apologise – did a poor job of making clear in the previous submission. We have amended the manuscript extensively to address these points throughout the results (p. 12) and again in the discussion (p. 28) to point out the limitations.

Toning down conclusions and stating caveats

We have extensively edited the manuscript to reflect how the quality of the analysis is dependent on the quality of the segmentation and tracking. We have adjusted the text (p. 28), to suggest alternate methods for image segmentation and tracking, that may perform better for more challenging image sets, and how these can be used alongside CellProfiler to generate the data required for Traject3d.

To identify distinct patterns of behaviour over time (trajectories) we need to compare tracked objects over the same period of time. We do this for 18,922 individual spheroids over 76h in Supplementary Figure 10. Despite this extremely large number of tracked spheroids analysed, this meant that we are unable, by definition, to include objects tracked for shorter periods of time. This could occur for many reasons, including splitting/merging, but also because an object touches the edge of – or leaves - the field of view. Therefore, rather than being 'unclean' data, the data that makes it through this quality control is robust and appropriate. Instead, the limitation of our approach is that we possibly underestimate even stronger heterogeneity due to objects that we have been unable to track extensively, therefore not making it into trajectory analysis.

We have amended the manuscript to explicitly emphasise these points on p. 28.

There is a larger consideration here, though. We say the following without having taken offense from the reviewer or without any shred of intention to do the same to the reviewer. Rather, we elaborate on the reviewer's point as it is excellent, informed and sits with precision at the heart of what we have tried to do. What the reviewer describes as 'unclean' data instead represents the lack of a 'clean' narrative. No amount of improved tracking will compensate for the fact that in 3D spheroid culture moving objects that merge or split isn't simply poor segmentation or tracking, it is a bona fide event and a reality for the experimentalist. Philosophically, and as highlighted in excellent recent articles in the Nature family (see below), "*the pressure to provide 'clean' narratives is harmful for the scientific endeavour.*" Instead, our approach is to provide capture of as many events as we can, however inconvenient to a clean narrative, and allow the user to interpret the data. The system may not be perfect but it is certainly, in our humble opinion, a step towards improvement compared to the biased counting a few spheroids by eye that is currently de rigueur.

Discussion of the problem with providing a 'clean' narrative:

<https://doi.org/10.1038/s41562-020-0818-9>

<https://doi.org/10.1038/s41562-021-01203-8>

5. (You really also should have in the supplemental material your whole western blots, with loading controls for EVERY blot not just representative ones!)

GAPDH loading controls were performed for every blot but not usually included. It is stated in figure legends that GAPDH is representative, including which blot it is for. Whole Western blots are now provided as Supplementary Figure 19.

Reviewer #3 (Remarks to the Author):

Freckmann and Sandilands et al describe the computational method Trajectory identification in 3D culture (Traject3d) to analyse live 3D cell models. The method uses the widely available Incucyte imaging system so could be widely adopted by researchers.

This reviewer is not qualified to critique the computational elements of the paper and leaves this area to other reviewers.

The authors train the method on several cell lines that appear to have been grown as planer (2.5D) clumps in Matrigel – which in my experience is quite an unusual cancer model (but maybe the authors can explain why they are relevant).

Typically, if researchers decide to use more complex 3D cultures over 2D cultures, it is to model a phenotype that can only be achieved in in 3D (e.g. self-organisation, differentiation, nutrient gradients, invasive migration etc). While the computational

imaging methods may be impressive, the lack of a more biomimetic cell culture model will limit the impact of the work.

The manuscript is well written but comprises lots of small observations that I struggled to collectively summarise once I had finished reading. The figures are nicely presented if slightly difficult to understand on first viewing.

Major Points

1) The 3D cultures used in the manuscript are extremely simple and lack much of the complexity of contemporary biomimetic models – notably spheroids and organoids. The cell lines may be appropriate for building the initial tools, but greatly limits the real-world use of Traject3d. Can the authors demonstrate the method working on a more commonly used 3D model such as spheroids or organoids? If the method can only work on clumped cell lines this will limit the application and should be clearly stated in the abstract. Currently the abstract specially mentions organoids but I could not find any evidence of them in the main text of the manuscript.

The exact nomenclature for 3D cultures can be confusing. Historically, the term used should relate to the anatomical origin of the cells. For prostate, these 3D culture would have been historically termed acini. We have edited the text (p. 6) to explicitly state that these structures are 3D spheroids, and to address the rationale behind using this 3D culture method.

We also include new data (Supplementary Fig 18) that shows the approach is equally applicable to the growth of organoids (in this instance, murine tumour-derived) in 3D domes. Therefore, our approach is likely applicable to any 3D approach that can be imaged over time.

We appreciate that, because of the wide field of view we provided for many figures, it may have appeared that we are looking at 'clumped cell lines' in '2.5D'. We now include maximum projections of optical sections of these from confocal imaging, making it easier to appreciate how these are not simply cell clumps, but rather highly polarised 3D structures (Supplementary Figure 5). Other examples of this can be seen in our recent work¹.

We respectfully disagree with the reviewer's assertion this approach is not 3D. This methodology for 3D culture has been used extensively for the past 3 decades for studies on polarity in vitro, including more recently to study polarisation in early embryo-like structures in vitro^{1-10, 11}. This is a tried and tested 3D approach.

Though Traject3d works with both domes and the sandwich method of 3D culture, the reduction in matrix to a thin coat on a dish, with cells plated on top, followed by matrix being overlaid generates 3D cultures surrounded by matrix is more cost-efficient in both generation of 3D structures (less matrix) and in image acquisition (keeping objects in a single plane). It is for this reason that we provided this method as a compatible way to scale up live imaging in parallel of spheroid cultures. We update the manuscript (p. 6) to include this information.

2) One of the primary uses of single-cell analysis is to resolve different cell types. Can Traject3d be used to perform cell-type-specific analysis of co-cultures? For example, in the PC3-EMT model, is data being produced for both PC3 cells and macrophages?

Demonstration of co-culture analysis would greatly increase the usefulness of the method.

This is a great question (!) and where we would eventually like to go with Traject3d. The answer is: yes and no.

The yes: If one wanted to compare different populations, for example 'normal' versus 'tumour', one could simply add different coloured labels (e.g. red vs green dyes, or genetic labelling with fluorescent protein) to cells separately before combining in 3D. This would allow one to distinguish the populations and ask if non-cell-autonomous behaviours were occurring. Some slight, achievable, tweaking of the CellProfiler pipelines would be needed to do this.

The no: the challenge is what happens if/when these populations physically interact (i.e. come into contact). This may be exactly what is wanted when, for instance, an immune cell contacts an organoid. The design of Traject3d is based on what happens to distinct objects in parallel in a given well. Several additional factors would need to be added to the approach to deal with this resulting data, not the least of which includes barriers in CellProfiler's approach (see response to Reviewer 2 regarding cells that touch). This is likely multiple years of additional work and validation, and therefore outside the scope of the current work.

In the PC3-EMT model, we clarify that these cells are only the prostate cells reisolated after co-culture (p. 9).

Our opinion is that while we agree that co-culture analysis is an excellent goal, it does not negate the extensive use of 3D mono-culture experiments wide-spread in many labs. Traject3d is aimed as providing a useful methodology for this type of commonly used approach and user.

References

1. Nacke, M. *et al.* An ARF GTPase module promoting invasion and metastasis through regulating phosphoinositide metabolism. *Nat. Commun.* 2021 121 **12**, 1–22 (2021).
2. Román-Fernández, Á. *et al.* The phospholipid PI(3,4)P₂ is an apical identity determinant. *Nat. Commun.* **9**, (2018).
3. Bryant, D. M. *et al.* A molecular network for de novo generation of the apical surface and lumen. *Nat. Cell Biol.* **12**, 1035–1045 (2010).
4. Bryant, D. M. *et al.* A molecular switch for the orientation of epithelial cell polarization. *Dev. Cell* **31**, 171–187 (2014).
5. MN, S. *et al.* Erratum: Pluripotent state transitions coordinate morphogenesis in mouse and human embryos. *Nature* **555**, 126 (2018).
6. Gálvez-Santisteban, M. *et al.* Synaptotagmin-like proteins control the formation of a single apical membrane domain in epithelial cells. *Nat. Cell Biol.* **14**, 838–849 (2012).
7. Martín-Belmonte, F. *et al.* PTEN-mediated apical segregation of phosphoinositides controls epithelial morphogenesis through Cdc42. *Cell* **128**, 383–397 (2007).
8. Martín-Belmonte, F. *et al.* Cell-polarity dynamics controls the mechanism of lumen formation in epithelial morphogenesis. *Curr. Biol.* **18**, 507–513 (2008).
9. Shahbazi, M. N. *et al.* Self-organization of the human embryo in the absence of maternal tissues. *Nat. Cell Biol.* **18**, 700–708 (2016).
10. Mangan, A. J. *et al.* Cingulin and actin mediate midbody-dependent apical lumen formation during polarization of epithelial cells. *Nat. Commun.* **7**, (2016).
11. Li, D., Mangan, A., Cicchini, L., Margolis, B. & Prekeris, R. FIP5 phosphorylation during mitosis regulates apical trafficking and lumenogenesis. *EMBO Rep.* **15**, 428–437 (2014).

Reviewer-Only Figure relating to Reviewer 1 point 4.

Heatmaps showing consistency of object frequency in data-driven **(a)** state and **(b)** trajectory classifications across biological replicates. Each column in the heatmaps is a biological replicate of the indicated sample, and the proportion (%) of spheroids in each classification is shown in purple to yellow. A chi-squared test was used to test the hypothesis that the biological replicate variable is independent of classification (state or trajectory). Due to high sample size, all tests were significant at $p < 0.01$. As such, Cramér's V statistic was calculated to determine the strength of the association, where values range from 0 (no association) to 1 (complete association). All values were < 0.5 , indicating weak, approaching no association between the classification and biological replicate variables.

This means that while there may be some slight variation, state and trajectory classifications are consistent across biological replicates.

Reviewers' Comments:

Reviewer #1:

Remarks to the Author:

The reviewer thank the authors for their efforts in addressing the comments. Though the technical description is clearer in the revised manuscript, this reviewer still has concerns with the proposed molecular mechanism. The author extensively characterizes the behaviors of PC3 derived cell lines models using their proposed methods and shows the different characteristics between epithelial-like vs mesenchymal-like PC-3 derived cell models using the proposed method. However, the underpinning molecular mechanism is still unclear. The authors propose ZEB1 - ESRP1/2 mutual inhibitory mechanism (Fig 4G, Supplementary Fig 16e) to explain the organoid behaviors association with EMT, which is not sufficiently demonstrated by their current data. The authors derive this idea through observing these two proteins' anti-correlated expression profiles, yet it's not clear if inhibition/activation of ZEB1 will lead to the activation/inhibition of ESRP1/2 or vice versa. The author only performs the knockdown of ZEB1 and ESRP1/2 on parental PC3 cells and characterizes the phenotype difference in the organoid model. The authors should also knockdown/knock-in in PC3-EPI and PC3-EMT pair as well as E-CAD+ and GS689.Li pair with ZEB1/ESRP1/2 and measure associated changes in protein expression and organoid phenotypes to see if epithelial-like phenotypes would transit from mesenchymal-like phenotypes or vice versa. This experiment can further clarify the regulatory role of ZEB1-ESRP1/2 in mediated EMT-associated phenotypes in the organoid.

In supplementary fig 16e, the authors propose HGF-c-Met-ZEB axis regulation that HGF can induce the dislocation of ZEB1 to the nucleus. However, from the author's own quantification, there seems to be minimal of difference in terms of effect size (Figures S16b) compared with or without HGF treatment, and perhaps some ZEB-1 stained images should be shown to support the analysis. ESRP1/2 expression should also be examined with HGF treatments to further support the proposed model.

Reviewer #2:

Remarks to the Author:

I applaud the authors for the work they did on this manuscript; it is much improved. Unfortunately, I am still not convinced the work is ready for publication, at least in this journal.

> The reviewer points out that multiple independent objects can have the same tracking ID if they touch at some point. What the reviewer refers to is the 'parent object' tracking ID, but each of these also has a 'child object' tracking ID.

No, this is incorrect. The value in my previous movie is the TrackObjects_Label_50, which is indeed the same value the authors are (sometimes, see below) filtering to only the largest.

> . This can be seen in the reviewer's example ('object 16', 'object 31', 'object 56', 'object 68', and 'object 88'). It is essential to emphasise here that these 'new' objects, as defined by having a new label, do not exist long enough to make it through these filtering steps we perform when identifying trajectories (see 'Trajectory Identification' on p. 41). Therefore, these are not affecting trajectory analysis.

Thank you for further clarifying exactly how your filtering is done; the re-assignment you have undertaken seems to control for some of what I was concerned about. I had not previously appreciated that you were essentially entirely re-assigning all track labels. Unfortunately this only partially solves the issue; as you can see in the diagram below (from Experiment 1_Plate 1_B7_2), this does not keep tracks from "skipping" back and forth across two split children based on which is larger at each moment. While I am sympathetic to the reviewers point is that we shouldn't let the perfect become the enemy of the good, this is a problem.

Ultimately though, the tool is really difficult to use, requiring a lot of different tools (CellProfiler, CellProfiler-Analyst KNIME, R, Conda), and a lot of different subpackages of many of those tools. I couldn't install some of the required R packages, because they are not supported on more up-to-

date versions of R and I did not want to downgrade my local machine. I still can't get KNIME to run on the authors' provided data - I've configured the path, but it refuses to load, I don't know if it's a Mac vs PC thing, a KNIME version thing, etc. Even if I had been able to load the data into KNIME, and provide the appropriate compute environment, if I were having issues with any of the metanodes, the manual is not detailed enough for me to troubleshoot many things nor is it clear to me where I could go for help or for how long a tool requiring such a large and fragile stack will end up being supported.

If the utility were more clear as to how users could interpret these classes and/or "do" with them once they were identified (assuming they are truly general), it MIGHT be worth the lift, but in its current state it's just too much difficulty to get working.

Reviewer #3:

Remarks to the Author:

The authors have addressed my comments.

Point-by-Point Rebuttal

Reviewer comments are presented in bold. Our responses, non-bold. In the manuscript and in our response below we highlight changes to the manuscript in blue. We number the reviewer points to allow for easier referencing of responses.

Reviewer#1

The reviewer thanks the authors for their efforts in addressing the comments. Though the technical description is clearer in the revised manuscript, this reviewer still has concerns with the proposed molecular mechanism.

We thank the reviewer for appreciating our clarification in the previous manuscript. We hope that our responses below address the remaining concerns.

1. The author extensively characterizes the behaviours of PC3 derived cell lines models using their proposed methods and shows the different characteristics between epithelial-like vs mesenchymal-like PC-3 derived cell models using the proposed method. However, the underpinning molecular mechanism is still unclear. The authors propose ZEB1 - ESRP1/2 mutual inhibitory mechanism (Fig 4G, Supplementary Fig 16e) to explain the organoid behaviours association with EMT, which is not sufficiently demonstrated by their current data. The authors derive this idea through observing these two proteins' anti-correlated expression profiles, yet it's not clear if inhibition/activation of ZEB1 will lead to the activation/inhibition of ESRP1/2 or vice versa. The author only performs the knockdown of ZEB1 and ESRP1/2 on parental PC3 cells and characterizes the phenotype difference in the organoid model. The authors should also knockdown/knock-in in PC3-EPI and PC3-EMT pair as well as E-CAD+ and GS689.Li pair with ZEB1/ESRP1/2 and measure associated changes in protein expression and organoid phenotypes to see if epithelial-like phenotypes would transit from mesenchymal-like phenotypes or vice versa. This experiment can further clarify the regulatory role of ZEB1-ESRP1/2 in mediated EMT-associated phenotypes in the organoid.

The interplay between ZEB1 and ESRP1/2 is well established in the literature wherein ESRP1/2 are direct transcriptional (repressed) targets of ZEB1. The reviewer is correct that in neither the literature nor our data is there a clear demonstration of ESRP1/2 conversely inhibiting ZEB1. We therefore removed suggestions of a mutual inhibition from the manuscript (removed Fig 4g and edited Fig S16e).

The manuscript's main focus is on providing the tools to identify that heterogeneity exists within 3D cultures, which happened to uncover that the well-studied ZEB1-ESRP1/2 EMT pathway is differentially regulating behaviours within these cultures. We add to this by demonstrating that heterogeneity in HGF signalling within a culture helps to drive this ZEB1 pathway. We argue that a deeper molecular dissection is outside of the main focus and point of our manuscript.

2. In supplementary fig 16e, the authors propose HGF-c-Met-ZEB axis regulation that HGF can induce the dislocation of ZEB1 to the nucleus. However, from the author's own quantification, there seems to be minimal of difference in terms of effect size (Figures S16b) compared with or without HGF treatment, and perhaps some ZEB-1 stained images should be shown to support the analysis. ESRP1/2 expression should also be examined with HGF treatments to further support the proposed model.

We respectfully suggest that the reviewer misunderstands. In Fig S16, spindle-state cells have both higher overall labelling for ZEB1 and a higher nuclear:cytoplasm ratio (i.e. more ZEB1 in the nucleus) at steady state (i.e. DMSO-treated cells). Indeed, the reviewer is correct that HGF treatment does only cause a modest (though significant) increase in the level of nuclear ZEB1 in 3D spheroids. However, the biologically significant effect is that HGF robustly increases the fraction of cells with spindle shape (spindle-type spheroid frequency increases from 22.5% in DMSO controls to 49.6% upon HGF treatment). The effect of HGF is therefore not to make already spindle cells have even more nuclear ZEB1, but rather to increase the proportion of spheroids with spindle features (which have the highest ZEB1 nuclear levels of all three shapes). We have re-written this section in the manuscript (highlighted in the revised manuscript in blue text) to make this point clearer.

“Notably, while the addition of HGF only caused a modest alteration to ZEB1 total expression and a modest but significant increase in ZEB1 nuclear translocation in Spindle-state spheroids, it robustly switched (from 22.5% in DMSO condition to 49% in HGF-treated condition) the majority of the population to Spindle state spheroids (which display high nuclear ZEB1) (Supplementary Figure 16b-d).”

We feel that the quantitation across multiple spheroids and replicate experiments is more appropriate than additional images from a single spheroid.

Reviewer#2

I applaud the authors for the work they did on this manuscript; it is much improved. Unfortunately, I am still not convinced the work is ready for publication, at least in this journal.

We appreciate the reviewer recognising an improvement in the last revision. We believe that there are some remaining misunderstandings that the reviewer has about our approach. This means we needed to do a better job in communicating our approach, which we take responsibility for. We hope to have corrected this below.

1. > The reviewer points out that multiple independent objects can have the same tracking ID if they touch at some point. What the reviewer refers to is the ‘parent object’ tracking ID, but each of these also has a ‘child object’ tracking ID.

No, this is incorrect. The value in my previous movie is the TrackObjects_Label_50, which is indeed the same value the authors are (sometimes, see below) filtering to only the largest.

> . This can be seen in the reviewer's example ('object 16', 'object 31', 'object 56', 'object 68', and 'object 88'). It is essential to emphasise here that these 'new' objects, as defined by having a new label, do not exist long enough to make it through these filtering steps we perform when identifying trajectories (see 'Trajectory Identification' on p. 41). Therefore, these are not affecting trajectory analysis.

Thank you for further clarifying exactly how your filtering is done; the re-assignment you have undertaken seems to control for some of what I was concerned about. I had not previously appreciated that you were essentially entirely re-assigning all track labels. Unfortunately this only partially solves the issue; as you can see in the diagram below (from Experiment 1_Plate 1_B7_2), this does not keep tracks from "skipping" back and forth across two split children based on which is larger at each moment. While I am sympathetic to the reviewers point is that we shouldn't let the perfect become the enemy of the good, this is a problem.

We apologise for the continued confusion. Our approach does ensure ongoing uniqueness of tracked objects where splitting has occurred. We have included a schema here to hopefully aid explanation.

We rely on 3 object labels, which when cross-referenced allows us to maintain uniqueness of a tracking label when a splitting event occurs:

- 1) an ObjectNumber (ON), in which every object within an image has a unique label. This can in theory change from frame to frame as it is not based on tracking.
- 2) the TrackObjects_Label (TOL), which is an ID that is maintained across the lifetime of a tracked object.
- 3) the ParentObjectNumber (PON), which links together two consecutive frames of a tracked object. The PON is the ON of the object in the immediately preceding frame that the tracked child object came from.

To demonstrate how the label reassignment we undertake works, we present here a hypothetical tracking event that has a splitting event in the third frame.

In the first frame the spheroid has a object number (ON) of 1 and a TOL of 44. In the second frame, the tracking label (TOL) is maintained, but the ON in each frame can change; in frame 2, for example purposes, while the TOL stays as 44, the ON becomes 3.

As a result of the object splitting in frame 3, we end up with two daughter objects with the same TOL as the parent throughout their lifetime. We needed to come up with an approach that restores uniqueness of the TOLs.

In our approach the larger of the two daughter objects gets to keep the TOL from the parent. We do this by reassigning the smaller child a new TOL only at the point of splitting (i.e. the next available unique TOL numerically; Step A in the diagram). For

instance, while the bigger object gets to keep TOL 44, the smaller object (object A) is assigned a new TOL, 45. This rectifies duplicate TOLs at the point in which an object splits, but the subsequent tracking events of these daughter objects still retain the original duplicate TOLs. We then needed an approach for updating the TOLs of tracking events succeeding the initial split. As such, subsequent to this splitting event, we then reassign the TOL based on referencing the parent object number (PON), not which object is larger. In frame 4, for example, Object A is updated to maintain a TOL of 45. We do this based on cross-referencing the PON unique ID of 7, which tells us that the TOL should be 45. Similarly, object B retains TOL 44, because that's what the parent object's TOL was (assigned because we looked up the PON).

Therefore, our approach does indeed ensure uniqueness from a splitting event onwards. This means that there is not a switching back and forth of tracking labels based on which daughter is bigger at a given point.

We also queried our data as to whether this reassignment meaningfully contributed to our data. Our findings are:

- Reassignment of split objects is infrequent, occurring in 0.5% of the tracked objects that contributed to trajectory identification (i.e. 93 of 18,922 spheroids).
- Exclusion of these tracked spheroids that possessed a reassignment event from the dataset did not affect subtype grouping or statistical significance.

Therefore, splitting events do not, in our datasets, meaningfully impact our findings. This may be the case in other datasets that we have not interrogated, so we believe that inclusion of this tracking label reassignment is both appropriate and robust.

2. Ultimately though, the tool is really difficult to use, requiring a lot of different tools (CellProfiler, CellProfiler-Analyst KNIME, R, Conda), and a lot of different subpackages of many of those tools. I couldn't install some of the required R packages, because they are not supported on more up-to-date versions of R and I did not want to downgrade my local machine. I still can't get KNIME to run on the authors' provided data - I've configured the path, but it refuses to load, I don't know if it's a Mac vs PC thing, a KNIME version thing, etc. Even if I had been able to load the data into KNIME, and provide the appropriate compute environment, if I were having issues with any of the metanodes, the manual is not detailed enough for me to troubleshoot many things nor is it clear to me where I could go for help or for how long a tool requiring such a large and fragile stack will end up being supported.

We are not sure why the reviewer cannot get the pipeline to run. We checked what we have provided, to make sure there wasn't simply an error in the examples. What we provided to the reviewers works in our hands. Here's what we did to ensure that things work:

- We re-downloaded the provided KNIME pipeline for Traject3d from GitHub
- We re-downloaded the provided sample data from the dropbox link on our GitHub.

- We loaded the data successfully on 4 different computers (a PC run as a virtual machine, a PC run as a Remote Desktop, locally on a PC, an iMac).
- 2 different authors independently did this.
- In all instances, the pipelines and the data loaded and ran.

We have extensively overhauled (see points below) both the KNIME pipeline and user manual to make Traject3d easier to navigate. We have uploaded these updated versions to the Traject3d GitHub.

KNIME Pipeline

- Updated and tested Traject3d pipeline for compatibility with latest version of R (4.2.0).
- Need for Python 2 environment removed so installation/integration with Anaconda easier. Unfortunately, a singular package (GeoSketch; ensures equal subsampling) does not exist solely in R, so we cannot remove Python requirement completely.
- Added simple instructions within the KNIME pipeline to show how to open, configure and execute nodes/components and metanodes.
- Coloured boxes used to separate pipeline workflow structure into functional groups to make it easier to follow i.e. Data Import and Processing, Feature Analysis, Data Driven Classification and User-Defined Classification.
- Coloured boxes added within metanodes to group nodes/components performing a specific function i.e. within User-Defined Classification metanode the nodes/components are grouped into Subsampling and tSNE, Outlines and Shape Classification Quantitation.
- Yellow and orange boxes added to indicate where node/components and metanodes require user configuration.
- Provided videos whereby we run our sample data through the Traject3d KNIME pipeline to show users how it should work (on Traject3d GitHub).

User Manual

- Added an introduction/overview on Traject3d pipeline and what it does.
- Updated and simplified instructions on KNIME installation and addition of extensions.
- Added clearer instructions for R Integration (latest version of R and Rserve extension) and clarification that latest version of R is compatible.
- Added clearer instructions for Python Integration and removed the need to add Python 2 environment.
- Added introductory and useful tip sections on how to use KNIME and supplied a link to more comprehensive instructions.
- For each node we have expanded the description to include a summary of what the node does, whether user configuration is required, instructions on how to configure and execute, stated what the output is and where it is saved. Where

decisions have to be made we have stated what we did for our data – e.g. what number of objects to select when subsampling for tSNE.

- Added Troubleshooting Section to suggest solutions for various problems that may arise when using Traject3d.

We also state here that we will of course aid those trying to use Traject3d. In addition to supplying test data, CellProfiler analysis pipelines, Traject3d pipelines and videos demonstrating how to run and load data on the Traject3d GitHub account, we have setup a dedicated query email (address available on Traject3d GitHub account) for this: query.traject3d@gmail.com

In addition, we also expect that interested parties might directly email the corresponding author (david.bryant@glasgow.ac.uk) if they required assistance.

We have a lot of experience in using aspects of Traject3d across multiple projects outside of what is presented in this manuscript (e.g. Sandilands, BioRxiv, <https://doi.org/10.1101/2022.04.25.489355>). We therefore have the experience in which multiple different users have been able to get Traject3d to work, and can help new users do the same. In full candour, however we cannot be IT support for those wanting to do Bioimage analysis or use KNIME for the first time. To get started, we provide an extensive user manual, which includes a step-by-step guide, a video demonstrating Traject3d in action, and a question section about what may be common errors. We will endeavour to provide genuine assistance to those that have first obtained local support from their IT to install Traject3d if needed. In the case of needing to do something that we have not tested for, this would need to occur as a discussion and potential collaboration.

Reviewer#3

The authors have addressed my comments.

We thank the reviewer for their assessment of our work.

ON: 1
TOL: 44
PON: -

3
44
1

(B)
2
44
44
2
11
44

ObjectNumber (ON)
TrackObjects_Label (TOL)
ParentObjectNumber (PON)

(A)
7
44 45
3
9
44 45
7

Reviewers' Comments:

Reviewer #1:

Remarks to the Author:

The reviewer still thinks the raised questions were not addressed. The discussed molecular mechanism in HGF-ZEB1-ESRP12 axis is not well supported by the provided data, mainly correlative observations and lack of casualty insights. Authors claim that the translocation of ZEB1 to nuclei is the key factor driving the HGF-induced organoid state change. The N/C ratio of ZEB1 appears to be only marginally increased in the organoids classified as spindles types in HGF treated group and not observed in the other two states though there is a statistical difference. The only observing the slight change in the N/C ratio of ZEB1 is not sufficient to explain the HGF-induced phenotype changes. The mechanistic model shown in supplementary figure 16 (A and E) is under-supported by the data. The authors should either perform the suggested experiments to develop deeper insight into the molecular pathways or rewrite the paragraphs with statements/plots that better fit and supported by the data.

Reviewer #2:

Remarks to the Author:

I thank the authors for their work to expand the documentation and to alter the claims in the paper to be more in line with the tool as it stands. I still think that if the authors are interested in this tool being usable by others, they should consider the sorts of parameters users are most likely going to want to change (object names, folder names, and the TrackObjects distance in pixels) and add either instructions on where to change them or investigate ways to parameterize them in the scripts (it seems like KNIME's String Manipulation node would allow it to be trivial to ie change TrackObjects_Label_50 to TrackObjects_Label_25, though I have not tested it). But I will no longer object to publication.

Point-by-Point Rebuttal

Reviewer comments are presented in bold. Our responses, non-bold. In the manuscript and in our response below we highlight changes to the manuscript in blue. We number the reviewer points to allow for easier referencing of responses.

Reviewer #1:

The reviewer still thinks the raised questions were not addressed. The discussed molecular mechanism in HGF-ZEB1-ESRP12 axis is not well supported by the provided data, mainly correlative observations and lack of casualty insights. Authors claim that the translocation of ZEB1 to nuclei is the key factor driving the HGF-induced organoid state change. The N/C ratio of ZEB1 appears to be only marginally increased in the organoids classified as spindles types in HGF treated group and not observed in the other two states though there is a statistical difference. The only observing the slight change in the N/C ratio of ZEB1 is not sufficient to explain the HGF-induced phenotype changes. The mechanistic model shown in supplementary figure 16 (A and E) is under-supported by the data. The authors should either perform the suggested experiments to develop deeper insight into the molecular pathways or rewrite the paragraphs with statements/plots that better fit and supported by the data.

We have toned down the statements made in the manuscript to reflect how the reviewer notes that nuclear translocation of ZEB1, while significantly altered in spindle cells, is a modest effect. We have extensively updated this section of the results and the discussion. These changes include the following:

We have rearranged the order of the panels in Supplementary Figure 16 and have rewritten the paragraph on page 21.

“Total ZEB1 expression, as well as the ratio of nuclear:cytoplasmic localisation, was modestly higher in spheroids displaying Spindle shape in comparison to Round and Spread spheroids both at steady state and upon HGF treatment (Supplementary Figure 16b, c). Notably, the addition of HGF only caused a modest alteration to ZEB1 total expression and a modest but significant increase in ZEB1 nuclear translocation in Spindle-state spheroids. However, HGF treatment robustly increased the proportion of Spindle-state spheroids in the population (from 22.5% in DMSO condition to 49%) (Supplementary Figure 16d). These data therefore suggest that the main effect of HGF is not to increase nuclear ZEB1 in spheroids already exhibiting Spindle state, but rather to increase the proportion of spheroids with Spindle features (which have the highest ZEB1 nuclear levels of all three shapes). However, further investigation would be required to confirm a direct causal effect of increased nuclear ZEB1 on Spindle shape and rule out the involvement of other mechanisms.”

We have removed the schema from Supplementary Figure 16 panel e.

We have toned down the conclusions relating to the mechanism in this experiment in discussion on page 26.

“One powerful feature of Traject3d is the ability to deconvolve bulk RNAseq to identify biologically important pathways controlling alternate phenotypes. As proof of application, we identified that HGF signalling is one pathway controlling an alternate phenotype.”

Reviewer #2:

I thank the authors for their work to expand the documentation and to alter the claims in the paper to be more in line with the tool as it stands.

We thank the reviewer for appreciating the improvements made to our documentation and manuscript clarity. We hope that the following comments will help to clear up any remaining uncertainty.

I still think that if the authors are interested in this tool being usable by others, they should consider the sorts of parameters users are most likely going to want to change (object names, folder names, and the TrackObjects distance in pixels) and add either instructions on where to change them or investigate ways to parameterize them in the scripts (it seems like KNIME's String Manipulation node would allow it to be trivial to ie change TrackObjects_Label_50 to TrackObjects_Label_25, though I have not tested it). But I will no longer object to publication.

Object names:

We provided information for the user to get around changes to the CellProfiler object names in the last revision. We apologise for not making this clearer previously.

Specifically, the easiest way for a user to get around this, rather than adapting the Traject3d KNIME pipeline, is to manually adjust the output file to match the expected filename. This can be found on page 4 of the Traject3d user manual (first bullet point, "Data"):

“If you adapt the provided CellProfiler pipeline and change the name of the objects being measured, this will be reflected in the naming of the output CSV file (i.e. no longer called “PhaseGrayCysts”). This will not be compatible with the provided version of Traject3d pipeline. An easy way to get around this is to rename your CellProfiler data output file to match the filename Traject3d expects (“PhaseGrayCysts.csv”). This avoids having to adapt Traject3d in order to accept a different input filename.”

Folder names:

No specific naming for an experiment is expected by Traject3d. However, the names of the experiment folder subdirectories (experimental replicate folders named "Experiment__n", and their own subdirectories "Data", "Experiment Key", "Phase", "PhaseGrayCystOutlines") are hardcoded. This means that the pipeline will not recognise these folders if the names are changed. We believe that the folder names are generic and unlikely to need changing by the user. Provision of functionality enabling a user to input custom names for, on average, 15 folders (one per experimental replicate and four respective subdirectories) will be more prone to confusion and error, potentially reducing ease of use in an attempt to make things more adaptable. Indeed, it is our opinion that the required directory structure will help with experimental clarity.

Related to the reviewer’s comment regarding changes to CellProfiler object names (see above), this could affect the name of the "PhaseGrayCystOutlines" subdirectory. We have expanded the user

manual to clarify how to get around changes to the naming of "PhaseGrayCysts", which can be found on page 5:

"As described above, changes to the CellProfiler object names may also affect the naming of this output folder. If this happens, the Traject3d KNIME pipeline will no longer be able to recognise the location of the outline images. The easiest way to rectify this is to manually edit the outlines subdirectory to ensure that it named as expected ("PhaseGrayCystOutlines")."

TrackObjects distance in pixels:

This had indeed been corrected in the previous submission. We apologise to the reviewer that this was not clearer in our previous response.

To summarise, our pipeline originally expected column names generated by CellProfiler when the TrackObjects distance was set to 50 pixels. The reviewer correctly pointed out that changes in this value would affect the resulting column names, meaning that the columns would no longer be recognised by our pipeline.

In our previous revision of the manuscript, pipeline and associated documentation, we adapted the pipeline to make it robust against changes to the TrackObject distance in pixels. Specifically, we adjusted the pipeline to identify the columns generated by the CellProfiler TrackObjects module and remove the final portion of the column names, which contains information relating to the set distance in pixels. This adjustment makes the column names generic (i.e. "TrackObjects_Label_25" and "TrackObjects_Label_50" are both converted into "TrackObjects_Label"). This means that if the user adjusts this distance in the CellProfiler TrackObjects module, our pipeline will now still recognise the output columns. As such, users will no longer have to adapt our provided Traject3d pipeline, making unnecessary the provision of instructions to do so. However, we have added clarification in the user manual to clarify this point (page 30):

"Problem: I changed the CellProfiler TrackObjects distance in pixels, and now the TrackObjects column names in my data file have also changed. Will this be a problem?"

Solution: This is fine! The Traject3d KNIME pipeline is robust to changes in this CellProfiler parameter so it will still recognise your TrackObjects columns."

Finally, we have added the following sentence to the manuscript (page 39) to direct reader to user manual for instructions on how to adapt the parameters.

"Some parameters in the CellProfiler and KNIME pipelines can be adapted for use with other datasets. For further information and instructions, refer to the associated user manual provided in the Traject3d GitHub repository."